# Heritability enrichment in context-specific regulatory networks improves phenotype-relevant tissue identification

**Zhanying Feng[1,2], Zhana Duren[3], Jingxue Xin[4], Qiuyue Yuan[3], Yaoxi He[5], Bing Su[5,6], Wing Hung Wong[4]\*, Yong Wang[1,2,6,7]\***

[1]CEMS, NCMIS, HCMS, MDIS, Academy of Mathematics and Systems Science, Chinese Academy of Sciences, Beijing, China; [2]School of Mathematics, University of Chinese Academy of Sciences, Chinese Academy of Sciences, Beijing, China; [3]Center for Human Genetics and Department of Genetics and Biochemistry, Clemson University, Greenwood, United States; [4]Department of Statistics, Department of Biomedical Data Science, Bio-X Program, Stanford University, Stanford, United States; [5]State Key Laboratory of Genetic Resources and Evolution, Kunming Institute of Zoology, Chinese Academy of Sciences, Kunming, China; [6]Center for Excellence in Animal Evolution and Genetics, Chinese Academy of Sciences, Kunming, China; [7]Key Laboratory of Systems Biology, Hangzhou Institute for Advanced Study, University of Chinese Academy of Sciences, Chinese Academy of Sciences, Hangzhou, China

**\*For correspondence:**
whwong@stanford.edu (WHungW);
ywang@amss.ac.cn (YW)

**Competing interest:** The authors declare that no competing interests exist.

**Abstract** Systems genetics holds the promise to decipher complex traits by interpreting their associated SNPs through gene regulatory networks derived from comprehensive multi-omics data of cell types, tissues, and organs. Here, we propose SpecVar to integrate paired chromatin accessibility and gene expression data into context-specific regulatory network atlas and regulatory categories, conduct heritability enrichment analysis with genome-wide association studies (GWAS) summary statistics, identify relevant tissues, and estimate relevance correlation to depict common genetic factors acting in the shared regulatory networks between traits. Our method improves power upon existing approaches by associating SNPs with context-specific regulatory elements to assess heritability enrichments and by explicitly prioritizing gene regulations underlying relevant tissues. Ablation studies, independent data validation, and comparison experiments with existing methods on GWAS of six phenotypes show that SpecVar can improve heritability enrichment, accurately detect relevant tissues, and reveal causal regulations. Furthermore, SpecVar correlates the relevance patterns for pairs of phenotypes and better reveals shared SNP-associated regulations of phenotypes than existing methods. Studying GWAS of 206 phenotypes in UK Biobank demonstrates that SpecVar leverages the context-specific regulatory network atlas to prioritize phenotypes' relevant tissues and shared heritability for biological and therapeutic insights. SpecVar provides a powerful way to interpret SNPs via context-specific regulatory networks and is available at https://github.com/AMSSwanglab/SpecVar, copy archived at swh:1:rev:cf27438d3f8245c34c357ec5f077528e6befe829.

## Editor's evaluation

In this article, the authors develop a method to identify potentially causal tissues and cell types for complex diseases by performing heritability enrichment estimation using information from gene regulatory networks. This article is of significant interest to geneticists and biologists interested in unraveling the molecular basis of disease. The key claims of the article are well supported by the data. The work has the potential to inform our understanding of the genetics of complex diseases.

## Introduction

Genome-wide association studies (GWAS) have gained a great success to identify thousands of genetic variants significantly associated with a variety of human complex phenotypes. Interpretation of those genetic variants holds the key to biological mechanism discovery and personalized medicine practice. However, this task is hindered by the genetic architecture that the heritability is distributed across SNPs of the whole genome with linkage disequilibrium, which cumulatively affect complex traits. By quantifying the contribution of true polygenic signal considering linkage disequilibrium, LD Score regression (LDSC) provides a widely appreciated method to estimate heritability (*Bulik-Sullivan et al., 2015b*) and genetic correlation (*Bulik-Sullivan et al., 2015a*) from GWAS summary statistics.

Another obstacle to genetic variant interpretation is that SNPs contribute to phenotype through gene regulatory networks in certain cellular contexts, that is, causal tissues or cell types. Those tissues are characterized by different types of epigenetic data, which give the active regions of the genome that interact with transcription factors (TFs) to regulate gene expression. Stratified LDSC extends LDSC and can estimate the partitioned heritability enrichment in the functional categories (*Finucane et al., 2015*). The categories can be nonspecific genome annotations (such as coding, UTR, promoter, and intronic regions) and context-specific regulatory regions called from chromatin data of different cell types, such as DNase-I hypersensitive sites from DNase-seq data, accessible peaks from ATAC-seq data, histone marker, or TF binding sites from ChIP-seq data (AAP and SAP). Using expression data, the functional categories can be alternatively constructed by the 100 kb windows around the transcribed regions of specifically expressed genes (SEGs) (*Finucane et al., 2018*). Essentially, these strategies summarize the high-dimensional SNP signals from the whole genome into partitioned heritability enrichments of functional categories and successfully identify relevant cellular tissues for many phenotypes (*Finucane et al., 2015*).

The rapid increase of multi-modal data resources, especially matched gene expression, chromatin states, and TF binding sites (i.e., measured on the same sample), offers an exciting opportunity to construct better functional categories for estimating context-specific heritability enrichment. One efficient way is to integrate large-scale epigenomic and transcriptomic data spanning diverse human contexts to infer regulatory networks (*Duren et al., 2017*). Those regulatory networks provide rich context-specific information and usually comprise TFs, regulatory elements (REs), and target genes (TGs). Recently, we developed the PECA2 model to infer regulatory network from paired expression and chromatin accessibility data (*Duren et al., 2017*; *Duren et al., 2020*). The inferred regulatory networks have been used to identify the master regulators in stem cell differentiation (*Li et al., 2019*) and interpret conserved regions for the nonmodel organisms (*Xin et al., 2020*). Noncoding genetic variants can be interpreted in the regulatory networks on how they cooperatively affect complex traits through gene regulation in certain tissues or cell types. For example, genetic variants in the regulatory network of cranial neural crest cells (CNCC) are elucidated on how they affect human facial morphology (*Feng et al., 2021*). Regulatory networks can help identify two kinds of relevant cell types to COVID19 severity (*Feng et al., 2022b*). RSS-NET utilizes gene regulatory networks of multiple contexts and shows better tissue enrichment estimation by decomposing the total effect of an SNP through TF-TG regulations (*Zhu et al., 2021*) and HiChIP RE-TG regulations (*Ma et al., 2022*). The phenotype-associated SNPs are revealed to be functional in a tissue- or cell-type-specific manner (*Westra and Franke, 2014*). The advances in constructing regulatory networks and interpreting genetic variants with regulatory networks enlighten us to (1) assemble a more comprehensive context-specific regulatory network atlas by using paired expression and accessibility data across diverse cellular contexts; (2) build context-specific regulatory categories by focusing on RE's specificity compared with other contexts; and (3) systematically identify enriched tissues or cell types, relevance correlation, and the underlying SN-associated regulations.

Specifically, we propose SpecVar to first leverage the publicly available paired expression and chromatin accessibility data in ENCODE and ROADMAP to systematically construct context-specific regulatory networks of 77 human contexts, which cover major cell types and germ layer lineages. This atlas serves as a valuable resource for genetic variants interpretation in multicellular contexts. SpecVar then constructs regulatory categories in the genome with this atlas, which can significantly improve the heritability enrichment. Based on the heritability enrichment and p-value in our regulatory categories, SpecVar defines the relevance score to give the context-specific representation of the GWAS. In this article, we use six well-studied phenotypes, large-scale facial morphology, and UKBB phenotypes

to show that, for a single phenotype, the relevance score of SpecVar can identify relevant tissues more efficiently; and for multiple phenotypes, SpecVar can reveal relevance correlation by common relevant tissues, and underlying shared SNP-associated regulations. Compared to the existing methods, SpecVar shows novelty in three aspects: (1) SpecVar integrates paired gene expression data and chromatin, which are two types of easily accessible data with rich information, into regulatory networks. The gene expression and chromatin accessibility in the regulatory network are complementary to each other to reveal the high-quality active regulatory elements and genes for certain tissues or cell types to interpret genetic variants; (2) SpecVar highlights the comparison with other contexts by specificity to narrow down the regulatory molecules; and (3) SpecVar is more interpretable because it can explain the relevance to tissues by SNP-associated regulatory networks and interpret phenotype correlation through common relevant tissues and shared SNP-associated regulatory network. These results show that SpecVar serves as a promising tool for post-GWAS analysis.

## Results

### Overview of the SpecVar method

SpecVar assembled a context-specific regulatory network atlas and built the context-specific representation (relevance score and SNP-associated regulatory network) of GWAS summary statistics based on heritability enrichment. *Figure 1* summarizes the major steps of SpecVar to construct the context-specific regulatory network atlas and regulatory categories, calculate heritability enrichment and SNP-associated regulatory network, and investigate interpretable relevant tissues and relevance correlation.

We first constructed regulatory networks of $M$ ($M$ = 77 in this work) contexts. Each network is represented by a set of relations between TF and RE and between RE and TG. The $M$ contexts included samples from all three germ layers, such as 'frontal cortex' (ectoderm), 'fetal thymus' (mesoderm), and 'body of pancreas' (endoderm), which ensured the wide coverage and system-level enrichment (*Figure 1—figure supplement 1*). The context-specific regulatory networks were extracted based on the specificity of REs compared with other contexts, considering the hierarchical relationship of $M$ contexts (Materials and methods, *Supplementary file 1a*). The REs in the $i$th context-specific regulatory network were pooled to form a regulatory category $C_i$ in the genome, which restricted the annotation to context-specific REs associated with active binding TFs and nearby regulated TGs (*Figure 1a*). Our atlas led to $M$ regulatory categories, $C_1, C_2, \ldots, C_M$ of SpecVar. Given GWAS summary statistics, the $M$ regulatory categories allowed partitioned heritability enrichment analysis by stratified LDSC. For a phenotype, stratified LDSC modeled genome-wide polygenic signals, partitioned SNPs into categories with different contributions for heritability, and considered SNP's linkage disequilibrium with the following polygenic model:

$$E\left(\chi_j^2\right) = N\sum_i \tau_i l\left(j, i\right) + Na + 1 \tag{1}$$

Here, $\chi_j^2$ is the marginal association of SNP $j$ from GWAS summary statistics; $N$ is the sample size; $l\left(j, i\right) = \sum_{k \in C_i} r_{jk}^2$ is the LD score of SNP $j$ in the $i$th regulatory category $C_i$, where $r_{jk}$ is the genotype correlation between SNP $j$ and SNP $k$ in population; $a$ measures the contribution of confounding biases; and $\tau_i$ represents the heritability enrichment of SNPs in $C_i$. Stratified LDSC estimated the p value $p_i$ for the heritability enrichment $\tau_i$ by block Jackknife (*Finucane et al., 2015*).

We defined the relevance score ($R_i$) of this phenotype to $i$th context (*Figure 1a*) as follows by combining the heritability enrichment and statistical significance (p-value):

$$R_i = \tau_i \cdot \left(-\log p_i\right) \tag{2}$$

The relevance score ($R$ score) provided a decision trade-off between the heritability enrichment and p-value resulting from a hypothesis test. It offered a robust means to rank and select relevant tissues for a given phenotype (*Xiao et al., 2014*).

Meanwhile, SpecVar associated SNPs with context-specific regulatory networks for biological interpretation. We defined the association score ($A$ score) to prioritize the REs by combining its regulatory strength and association significance with the phenotype (averaged -log(p-value) of SNPs located

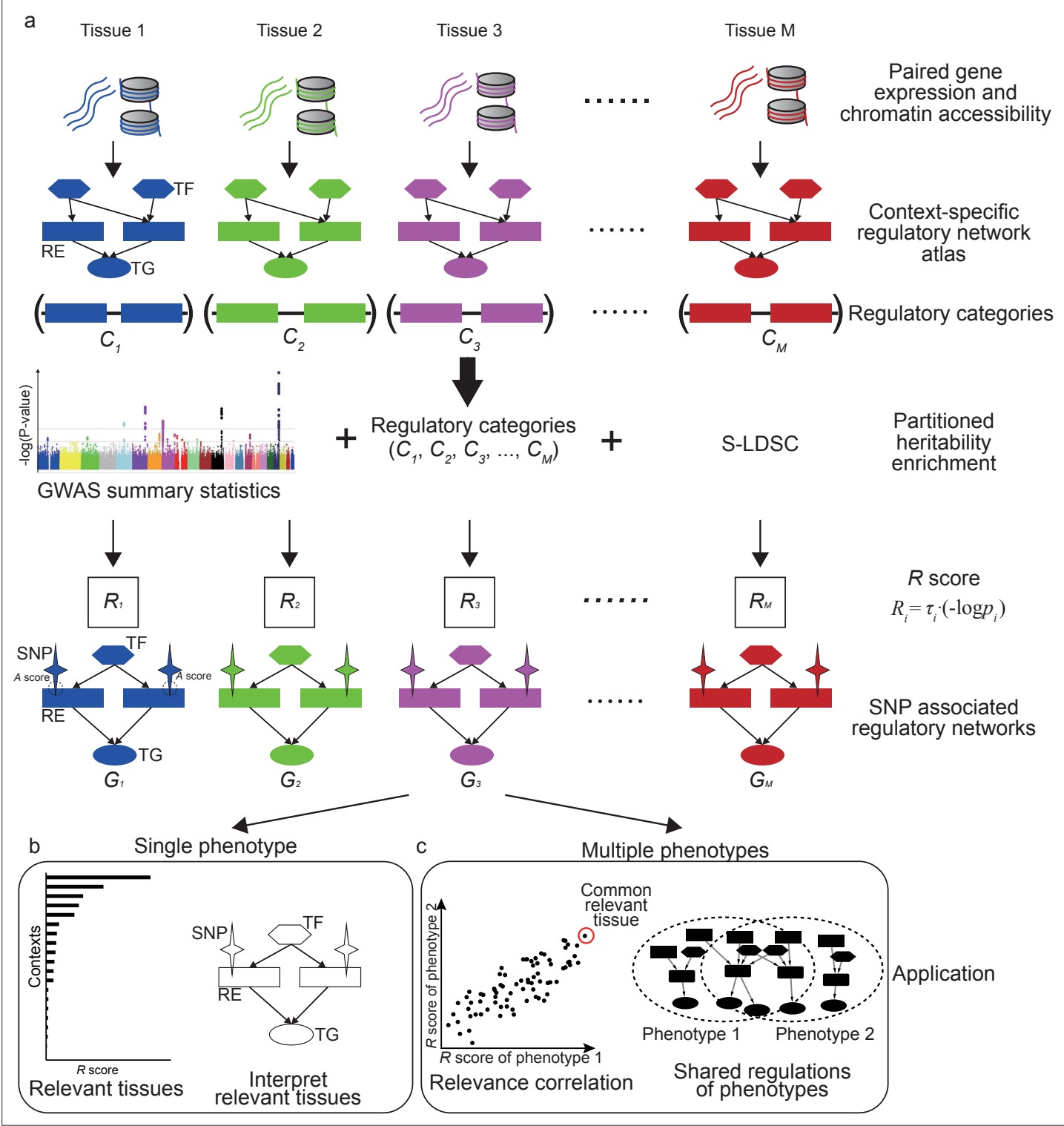

**Figure 1.** Overview of SpecVar. (**a**) SpecVar constructs an atlas of context-specific regulatory networks and regulatory categories. Then SpecVar represents genome-wide association studies (GWAS) summary statistics into relevance scores and SNP-associated regulatory subnetworks. (**b**) For a single phenotype, SpecVar can use relevance score and SNP-associated regulatory subnetworks to identify and interpret relevant tissues. (**c**) For multiple phenotypes, based on relevance score, SpecVar can reveal relevance correlation, common relevant tissues, and shared regulations.

The online version of this article includes the following figure supplement(s) for figure 1:

**Figure supplement 1.** Principal component analysis (PCA) plot of regulatory network atlas of 77 human tissues.

near the RE and downweighted by their LD scores and distances to this RE). We extracted the REs with significant $A$ scores ($FDR \leq 0.05$), as well as their directly linked upstream TFs, downstream TGs, and associated SNPs, to form the SNP-associated regulatory subnetwork (*Figure 1a*, Materials and methods). Given GWAS summary statistics of a phenotype, SpecVar obtained $M$ SNP-associated regulatory subnetworks, $G_1, G_2, \ldots, G_M$, allowing us to interpret relevant tissues by SNP's regulation mechanism.

The relevance score to diverse human contexts and SNP-associated regulatory networks allowed SpecVar to perform post-GWAS analysis. For a single phenotype, the $R$ scores indicated the relevance of this phenotype to $M$ contexts, which can be used to identify relevant tissues. Then in the relevant tissues, we can investigate the SNP-associated regulatory subnetwork to interpret the relevance (*Figure 1b*, Materials and methods). For multiple phenotypes, we can correlate the $R$ score vectors in multiple contexts to define relevance correlation (*Finucane et al., 2018*). The relevance correlation might give insights into the association of phenotypes since SpecVar can further interpret the relevance correlation between two phenotypes by common relevant tissues and the shared SNP-associated regulatory subnetwork (*Figure 1c*, Materials and methods).

## Context-specific regulatory networks improve heritability enrichment

We first designed experiments to show that context-specific regulatory networks could improve heritability enrichment. We collected GWAS summary statistics of six phenotypes, including two lipid phenotypes (*Willer et al., 2013*): low-density lipoprotein (LDL) and total cholesterol; two human intelligential phenotypes (*Lee et al., 2018*): educational attainment and cognitive performance; and two craniofacial bone phenotypes: brain shape (*Naqvi et al., 2021*) and facial morphology (*Xiong et al., 2019*). We used these six phenotypes as a benchmark since their relevant tissues have been previously studied and partially known: lipid phenotypes are associated with the liver for its key role in lipid metabolism (*Nguyen et al., 2008*); human intelligential phenotypes are associated with brain tissues (*Goriounova and Mansvelder, 2019*); facial morphology and brain shape have shared heritability in cranial neural crest cells (*Naqvi et al., 2021*). We compared our context-specific regulatory networks with four alternative methods of functional categories: all regulatory elements (ARE), all accessible peaks (AAP), specifically accessible peaks (SAP) (*Finucane et al., 2015*), and specifically expressed genes (*Finucane et al., 2018*) (SEG) (Materials and methods).

First, we showed that SpecVar could achieve higher heritability enrichment in the relevant tissues than other methods (*Supplementary file 1b*). For LDL, SpecVar obtained a heritability enrichment of 678.91 in the 'right lobe of liver', while ARE, SAP, AAP, and SEG gave heritability enrichment of 113.34, –42.09, 50.95, and 4.47, respectively. We conducted Welch's $t$-test to assess the significance of the difference between SpecVar and other methods and found that the heritability enrichment of SpecVar was significantly higher than ARE ($p = 6.9 \times 10^{-4}$), SAP ($p = 1.4 \times 10^{-4}$), AAP ($p = 3.4 \times 10^{-4}$), and SEG ($p = 2.1 \times 10^{-4}$) (*Figure 2a*). For total cholesterol, SpecVar also gave significantly higher heritability enrichment in 'right lobe of liver' than ARE ($p = 5.7 \times 10^{-4}$), SAP ($p = 4.4 \times 10^{-5}$), AAP ($p = 1.6 \times 10^{-4}$), and SEG ($p = 7.7 \times 10^{-5}$) (*Figure 2b*). For educational attainment and cognitive performance, they were relevant to brain tissues: 'frontal cortex', 'cerebellum', 'caudate nucleus', 'Ammon's horn', and 'putamen'. SpecVar obtained the highest averaged heritability enrichment in brain tissues among these methods (*Figure 2—figure supplement 1a and b*). In the 'frontal cortex', SpecVar had significantly higher heritability enrichment than ARE ($p = 1.2 \times 10^{-5}$), SAP ($p = 2.0 \times 10^{-6}$), AAP ($p = 3.0 \times 10^{-6}$), and SEG ($p = 3.0 \times 10^{-6}$) for educational attainment (*Figure 2c*). And for cognitive performance in 'frontal cortex', SpecVar also had significantly higher heritability enrichment than ARE ($p = 9.0 \times 10^{-6}$), SAP ($p = 2.0 \times 10^{-6}$), AAP ($p = 1.0 \times 10^{-6}$), and SEG ($p = 1.0 \times 10^{-6}$) (*Figure 2d*). For brain shape, SpecVar obtained a significantly higher heritability enrichment in its relevant context 'CNCC' than the other four methods (ARE $p = 5.9 \times 10^{-4}$, SAP $p = 6.7 \times 10^{-4}$, AAP $p = 7.5 \times 10^{-4}$, and SEG $p = 8.1 \times 10^{-5}$, *Figure 2e*). For facial morphology, SpecVar also gave a much higher heritability enrichment in 'CNCC' than the other four methods (ARE $p = 9.0 \times 10^{-6}$, SAP $p = 1.0 \times 10^{-6}$, AAP $p = 7.0 \times 10^{-6}$, and SEG $p = 1.0 \times 10^{-6}$, *Figure 2e*). Second, except for the known relevant tissues, these complex traits may be relevant to other contexts. So, for every method, we ranked the heritability enrichment to get the top 10 contexts and used these top contexts' heritability enrichment to compare the ability of these five methods to explain heritability. SpecVar also showed the best performance of heritability enrichment among the five methods (*Figure 2g*). Taking brain shape as an

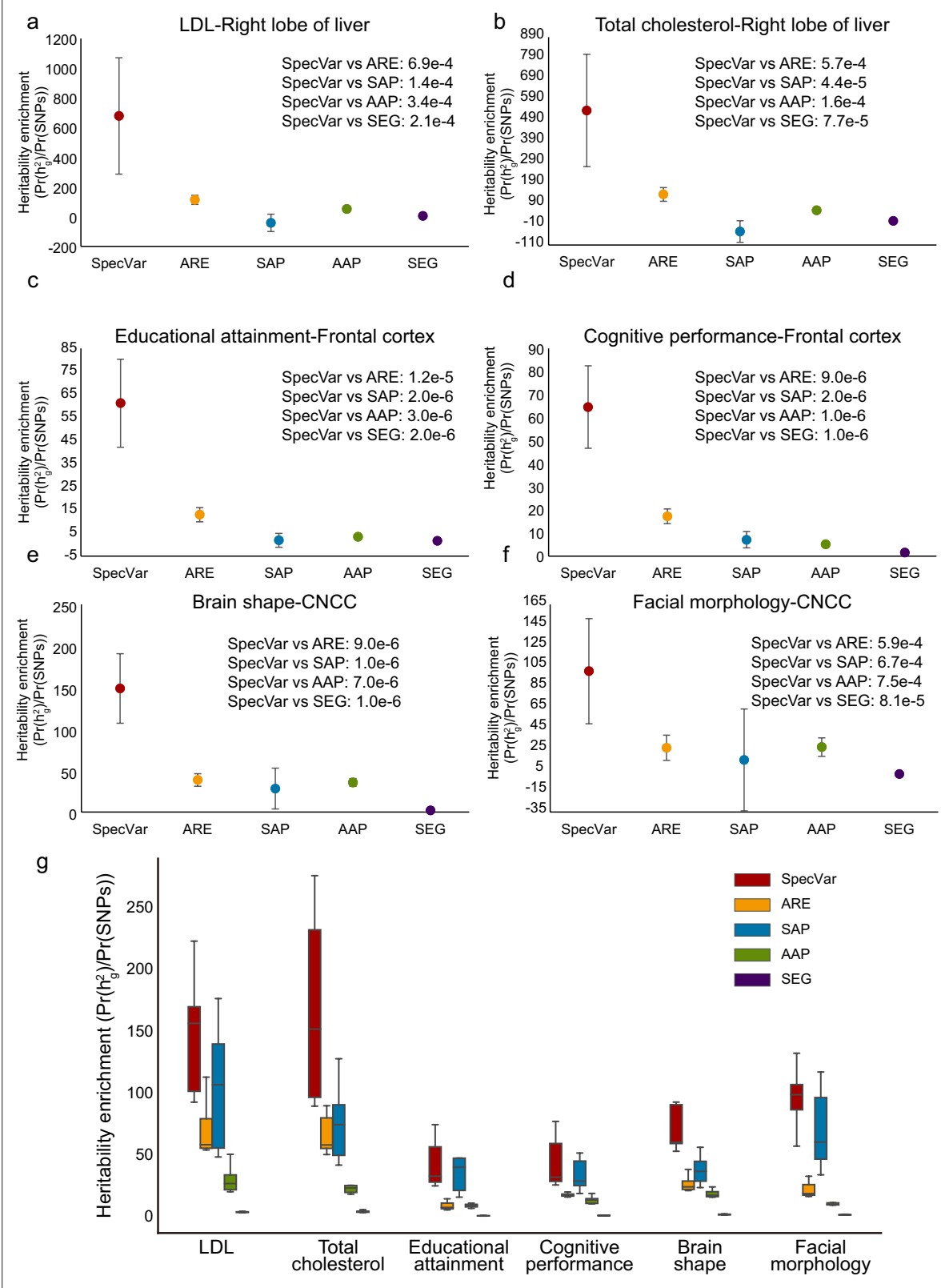

**Figure 2.** Comparison of heritability enrichment between SpecVar and four alternate methods: all regulatory elements (ARE), all accessible peaks (AAP), specifically accessible peaks (SAP), and specifically expressed genes (SEG). (**a**) The heritability enrichment of low-density lipoprotein (LDL) in the 'right lobe of liver'. (**b**) The heritability enrichment of total cholesterol in the 'right lobe of liver'. (**c**) The heritability enrichment of educational attainment in the 'frontal cortex'. (**d**) The heritability enrichment of cognitive performance in the 'frontal cortex'. (**e**) The heritability enrichment of brain shape in cranial

*Figure 2 continued on next page*

*Figure 2 continued*

neural crest cell (CNCC). (**f**) The heritability enrichment of facial morphology in 'CNCC'. The sample size of error bars for (a-f) is 200. (**g**) Boxplot of top 10 contexts' heritability enrichment of six phenotypes for five methods.

The online version of this article includes the following figure supplement(s) for figure 2:

**Figure supplement 1.** SpecVar achieves higher heritability than other methods through regulatory network and specificity.

**Figure supplement 2.** The heritability enrichment estimated by 'pooled genome partition'.

example, SpecVar achieved significantly higher heritability enrichment (averaged heritability enrichment 96.13) than ARE (averaged heritability enrichment 26.77, *t*-test $p = 3.4 \times 10^{-3}$), SAP (42.92, $p = 1.9 \times 10^{-2}$), AAP (20.34, $p = 1.8 \times 10^{-3}$), and SEG (2.25, $p = 3.1 \times 10^{-4}$). We found that specificity could significantly improve the heritability enrichment. Among the five methods we compared, SpecVar and SAP are based on the specificity of ARE and AAP, respectively. SpecVar showed significantly higher heritability enrichment than ARE and SAP showed significantly higher heritability enrichment than AAP (*Figure 2g*). For brain shape, SpecVar obtained averaged heritability enrichment of 96.31 of the top 10 contexts, which was significantly higher than ARE (averaged heritability enrichment 26.77, $p = 3.4 \times 10^{-3}$); SAP obtained average heritability enrichment of 42.92, and AAP's averaged heritability enrichment was 20.34 ($p = 2.7 \times 10^{-3}$). The other five phenotypes showed a similar improvement (*Figure 2g*).

To explore the heritability enrichment improvement of SpecVar, we conducted ablation analysis to study the contribution of two important parts of SpecVar: (1) regulatory network by integrating gene expression and chromatin accessibility data, and (2) specificity by comparing with other contexts. *Figure 2—figure supplement 1c* shows the relationship and difference of the five methods: SEG is the combination of gene expression and specificity; AAP is only from chromatin accessibility; SAP is the combination of chromatin accessibility and specificity; ARE integrates gene expression and chromatin accessibility; and SpecVar considers integration of gene expression, chromatin accessibility, and specificity. To analyze the effect of the regulatory network in heritability enrichment, we compared SpecVar with SAP and showed the effect of gene expression data. We compared SpecVar with SEG and showed the effect of chromatin accessibility data. We compared SpecVar with ARE and showed the contribution of specificity (*Figure 2—figure supplement 1d*). To quantify the effect of each component, we caculated the fold change of different methods' heritability enrichments. We found that chromatin accessibility, which was part of the regulatory network, showed the highest effect in improving heritability enrichments for all six phenotypes. This is consistent with the fact that most genetic variants are located in the noncoding regulatory regions (*Claussnitzer et al., 2015*; *Kumar et al., 2012*; *Smemo et al., 2014*) and chromatin accessibility gives the direct functional evidence for genetic variants. The specificity in SpecVar also contributed at least four-fold improvement in heritability enrichment (*Figure 2—figure supplement 1e*).

In summary, the experiment on six phenotypes' GWAS summary statistics proved that SpecVar achieved the best performance in explaining the heritability of phenotypes. These results demonstrated the power of integrating expression and chromatin accessibility data and considering contexts' specificity.

## SpecVar can accurately reveal relevant tissues for phenotypes

After establishing that SpecVar could use the context-specific regulatory networks to improve heritability enrichment, we next showed that for a given phenotype, SpecVar could use $R$ scores to identify relevant tissues more accurately than other methods. In this experiment, we also used the above six phenotypes with their known relevant tissues as a benchmark and first compared SpecVar with the other two specificity-based methods: SAP and SEG (Materials and methods).

For two lipid phenotypes, SpecVar revealed that both LDL and total cholesterol were most relevant to the 'right lobe of liver' (*Figure 3a and b*, *Table 1*), which was consistent with the existing reports that the liver plays a central role in lipid metabolism, serving as the center for lipoprotein uptake, formation, and export to the circulation (*Jha et al., 2018*; *Nguyen et al., 2008*). SpecVar found that LDL and the total cholesterol were also significantly relevant to the 'fetal adrenal gland' and the adrenal cortex has been revealed to play an important role in lipid mentalism (*Boyd et al., 1983*). However, SAP and SEG failed to prioritize liver tissue as the significant relevant tissue. For LDL, SAP

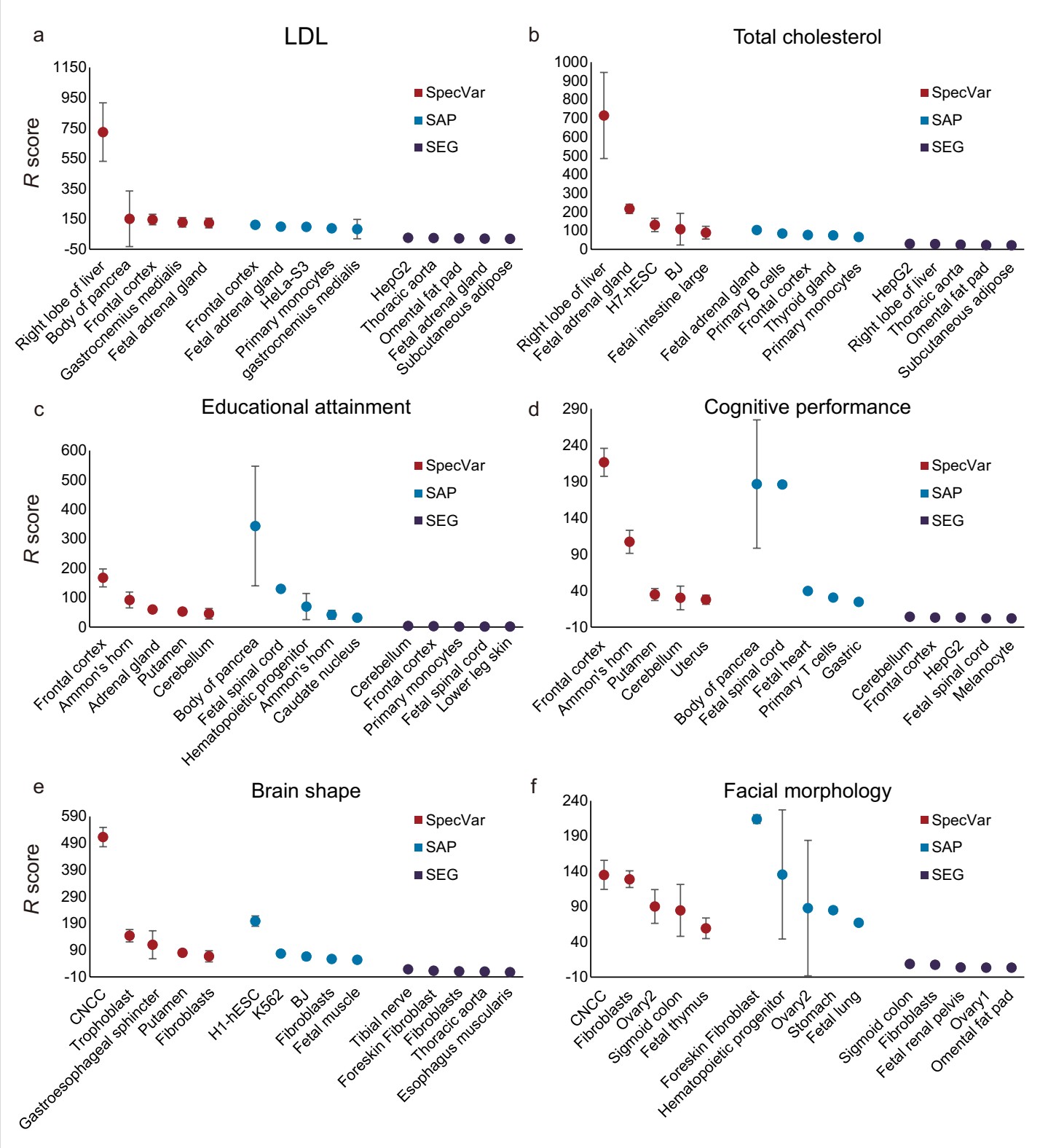

**Figure 3.** Comparison of identifying proper relevant tissues between SpecVar and two other LD Score regression (LDSC)-based method: specifically accessible peaks (SAP) and specifically expressed genes (SEG). The top five relevant tissues ranked by the relevant score of SpecVar, SAP, and SEG for (**a**) low-density lipoprotein (LDL), (**b**) total cholesterol, (**c**) educational attainment, (**d**) cognitive performance, (**e**) brain shape, and (**f**) facial morphology. The sample size of error bars for (a-f) is 100.

*Figure 3 continued on next page*

*Figure 3 continued*

The online version of this article includes the following figure supplement(s) for figure 3:

**Figure supplement 1.** The top five relevant tissues ranked by the log-likelihood estimated by CoCoNet for (**a**) low-density lipoprotein (LDL), (**b**) total cholesterol, (**c**) educational attainment, (**d**) cognitive performance, (**e**) brain shape, and (**f**) facial morphology.

**Figure supplement 2.** The top five relevant tissues ranked by -log(p-value) estimated by RolyPoly for (**a**) low-density lipoprotein (LDL), (**b**) total cholesterol, (**c**) educational attainment, (**d**) cognitive performance, and (**e**) facial morphology.

**Figure supplement 3.** Top 10 contexts ranked by heritability enrichment in context-specific regulatory elements of (**a**) low-density lipoprotein (LDL), (**b**) total cholesterol, (**c**) educational attainment, (**d**) cognitive performance, (**e**) brain shape, and (**f**) facial morphology.

**Figure supplement 4.** Top 10 contexts ranked by p-values of heritability enrichment in context-specific regulatory elements of (**a**) low-density lipoprotein (LDL), (**b**) total cholesterol, (**c**) educational attainment, (**d**) cognitive performance, (**e**) brain shape, and (**f**) facial morphology.

**Figure supplement 5.** The top five relevant tissues ranked by the relevant score estimated by the group-based SpecVar.

identified the 'frontal cortex' to be the most relevant tissue. SEG identified the most relevant tissue to be 'HepG2', which was human hepatoma cell lines, but the relevance score was relatively lower (*Figure 3a*, *Supplementary file 1c*). For total cholesterol, SAP identified the 'fetal adrenal gland' and SEG obtained 'HepG2' as the most relevant tissues (*Figure 3b*, *Supplementary file 1c*).

For two human intelligential phenotypes, SpecVar prioritized the 'frontal cortex' to be the most relevant tissue for both educational attainment and cognitive performance (*Figure 3c and d*, *Table 1*). 'Frontal cortex' is the cerebral cortex covering the front part of the frontal lobe and is implicated in planning complex cognitive behavior, personality expression, decision-making, and moderating social behavior (*Gabrieli et al., 1998*; *Yang and Raine, 2009*). There were five tissues ('frontal cortex', 'Ammon's horn', 'cerebellum', 'putamen', and 'caudate nucleus') from the brain in our atlas and they were significantly higher ranked by SpecVar's relevance score than nonbrain tissues for educational attainment (Wilcoxon rank-sum test, $p = 6.1 \times 10^{-7}$ , *Figure 3c*) and cognitive performance ($p = 8.0 \times 10^{-6}$, *Figure 3d*). In comparison, for educational attainment, SAP prioritized brain tissues to be higher ranked than nonbrain tissues, but with a less significant p-value ($p = 2.3 \times 10^{-3}$, *Figure 3c*, *Supplementary file 1c*). SEG could not rank brain tissues to be higher than nonbrain tissues ($p = 6.4 \times 10^{-1}$, *Figure 3c*, *Supplementary file 1c*). For cognitive performance, SAP failed to rank brain tissues as the more relevant tissues ($p = 6.0 \times 10^{-2}$, *Figure 3d*, *Supplementary file 1c*), and SEG identified brain tissues to be more relevant than nonbrain tissues but with a less significant p-value ($P = 3.2 \times 10^{-3}$, *Figure 3d*, *Supplementary file 1c*).

For both facial morphology and brain shape, SpecVar identified CNCC as the most relevant context (*Figure 3e and f*, *Table 1*). CNCC is a migratory cell population in early human craniofacial

**Table 1.** The total sample size, number of significant SNP associations, and SpecVar-identified relevant tissues of six phenotypes.

For each relevant tissue, we have two numbers in the bracket: the first is the *R* score and the second is its false discovery rate (FDR) q-value.

| Trait | Sample size | SNP association | Relevant tissues (*R* score and its FDR q-value) |
| --- | --- | --- | --- |
| Low-density lipoprotein | 173,082 | 3077 | Right lobe of liver (722.74, 1.2e-3), frontal cortex (146.54, 3.3e-4), gastrocnemius medialis (128.02, 4.7e-4), fetal adrenal gland (123.52, 9.5e-4) |
| Total cholesterol | 187,365 | 4169 | Right lobe of liver (714.75, 1.0e-2), fetal adrenal gland (216.74, 4.3e-17), H7-hESC (130.73, 2.1e-3) |
| Educational attainment | 1070,751 | 30,519 | Frontal cortex (167.23, 3.7e-7) |
| Cognitive performance | 257,841 | 13,732 | Frontal cortex (216.62, 7.0e-28), Ammon's horn (107.25, 2.3e-10) |
| Brain shape | 19,644 | 38,630 | CNCC (512.56, 2.7e-44), trophoblast (144.15, 3.8e-9) |
| Facial morphology | 10,115 | 495 | CNCC (134.95, 8.0e-10), fibroblast (128.81, 3.8e-26) |

CNCC, cranial neural crest cell.

development that gives rise to the peripheral nervous system and many non-neural tissues such as smooth muscle cells, pigment cells of the skin, and craniofacial bones, which make it much more related to facial morphology and brain shape than the other 76 contexts (*Cordero et al., 2011*; *Barlow et al., 2008*). Facial morphology and brain shape were also revealed to share heritability in CNCC (*Naqvi et al., 2021*). But the other two methods failed to identify CNCC as the most relevant context. For brain shape, SAP identified 'H1-hESC' and SEG identified 'tibial nerve' to be the most relevant tissue (*Figure 3e*, *Supplementary file 1c*). For facial morphology, SAP and SEG identified 'foreskin' and 'sigmoid colon' to be the most relevant tissues, respectively (*Figure 3f*, *Supplementary file 1c*).

We next compared with other relevant tissue identification methods that were not based on LDSC. First, we compared SpecVar to CoCoNet, which was based on gene co-expression networks. CoCoNet is built with 38 tissues' co-expression networks from GTEx, and we applied it to our six phenotypes. We could see CoCoNet identified 'Breast' as the most relevant tissue for LDL (*Figure 3—figure supplement 1a*) and 'Brain_other' as the most relevant tissue for total cholesterol (*Figure 3—figure supplement 1b*). 'Breast' was the most relevant tissue for educational attainment (*Figure 3—figure supplement 1c*), and 'Stomach' was the most relevant tissue for cognitive performance (*Figure 3—figure supplement 1d*). Since there is no CNCC sample in GTEx, CoCoNet revealed 'Prostate' as the most relevant tissue for brain shape and facial morphology. These results seemed less reasonable than SpecVar because CoCoNet did not identify liver tissues for LDL and total cholesterol and did not reveal brain tissues for educational attainment and cognitive performance. We also compared with RolyPoly, which was a non-network-based method for discovering relevant tissues. We fitted the RolyPoly model with gene expression profiles of our 77 human contexts and applied it to GWAS of LDL, total cholesterol, educational attainment, cognitive performance, and facial morphology. RolyPoly prioritized the 'HepG2' cell line as the most relevant tissue for LDL and total cholesterol (*Figure 3—figure supplement 2a and b*). 'HepG2' is also reasonable to be relevant to lipid phenotypes because it is the nontumorigenic cell with high proliferation rates and epithelial-like morphology that performs many differentiated hepatic functions. For educational attainment, RolyPoly did not identify the five brain tissues as the top-ranked tissue and only include 'fetal spinal cord' in the top five relevant tissues (*Figure 3—figure supplement 2c*). For cognitive performance, there were no brain tissues in the top five tissues (*Figure 3—figure supplement 2d*). And for facial morphology, RolyPoly failed to identify 'CNCC' as relevant tissues (*Figure 3—figure supplement 2e*). The comparison with two non-LDSC-based methods again showed the superiority of SpecVar to identify proper relevant tissues.

After identifying the relevant tissues, SpecVar could further interpret the relevance by extracting SNP-associated regulatory subnetwork (Materials and methods). For example, we obtained the brain shape's SNP-associated regulatory subnetwork in CNCC (*Figure 4a*). There were 62 SNPs associated with 24 REs, 73 TFs, and 52 TGs. The TGs were tightly involved with brain development. For example, *POU3F3* is a well-known TF involved in the development of the central nervous system and is related to many neurodevelopmental disorders (*Snijders Blok et al., 2019*). *EMX2* is expressed in the developing cerebral cortex and is involved in the patterning of the rostral brain (*Cecchi and Boncinelli, 2000*). *FOXC2* is a member of the FOX family, which are modular competency factors for facial cartilage (*Xu et al., 2018*), and its mutation is linked to the cleft palate (*Bahuau et al., 2002*). By GWAS study, *FOXC2* was previously found to be associated with brain shape by its nearest significant SNP '16:86714715' (*Naqvi et al., 2021*). However, in CNCC, we did not find any accessible peaks that overlapped with this SNP. Instead, we found a CNCC-specific RE that regulated *FOXC2* in a locus of the 650k downstream (*Figure 4b*). GWAS revealed that the SNPs in this region had a strong association with brain shape and had high LD with each other. Our CNCC-specific regulations further prioritized only two SNPs ('16:87237097', '16:87236947') located in this CNCC-specific RE, which may influence the expression of *FOXC2* and the brain shape phenotypes. We checked the chromatin loops in the database of HiChIP (*Zeng et al., 2022*) and found that this SNP-associated regulation of *FOXC2* was supported by a HiChIP loop in brain tissues to link this SNP locus and *FOXC2* promoter. This example showed the power of SpecVar to interpret the genetic variants' association with phenotypes by detailed regulatory networks in relevant tissues.

The SNP-associated regulatory networks can facilitate the fine mapping of GWAS signals. Since there are eQTL data of the liver and brain tissues in the GTEx database, we collected the significant SNP-gene pairs of liver and brain tissues from GTEx to validate the identified SNP-associated

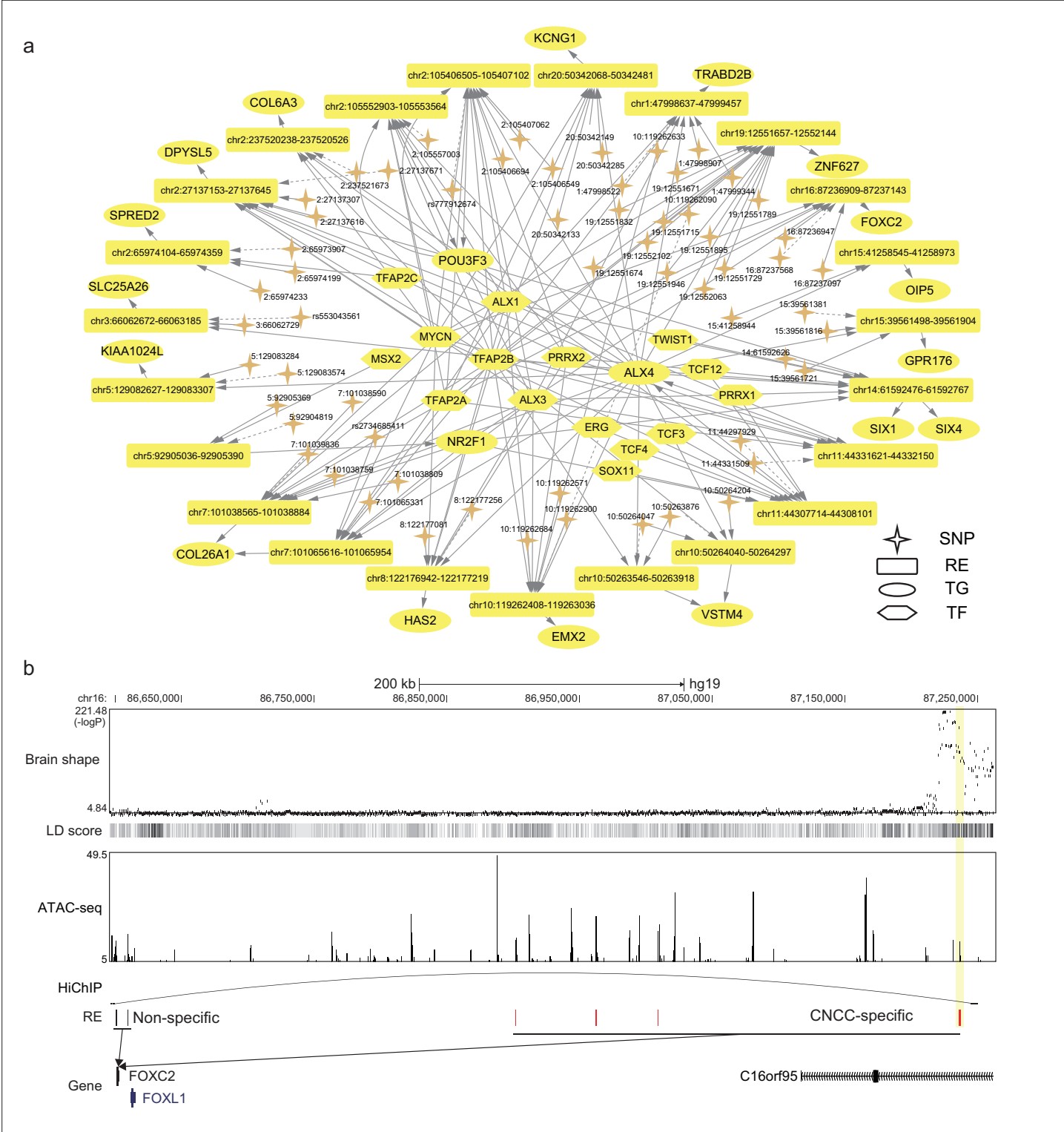

**Figure 4.** SpecVar uses SNP-associated regulation to interpret relevance to tissues. (**a**) The brain shape's SNP-associated regulatory subnetwork in cranial neural crest cell (CNCC). The dash arrows indicate significant SNPs that are not located in regulatory elements (RE) but near this RE. (**b**) SNP-associated regulation of *FOXC2*. There is a group of significant SNPs of brain shape that is located in the 650k downstream of *FOXC2*, and they are with high linkage disequilibrium. SpecVar prioritizes SNPs located in a CNCC-specific RE as causal genetic variants affecting brain shape through the regulation of *FOXC2*. The SNP locus and promoter of *FOXC2* are linked by a HiChIP loop of the brain tissues.

The online version of this article includes the following figure supplement(s) for figure 4:

*Figure 4 continued on next page*

*Figure 4 continued*

**Figure supplement 1.** Distribution of *A* scores of (**a**) low-density lipoprotein (LDL) in 'right lobe of liver', (**b**) total cholesterol in the 'right lobe of liver', (**c**) educational attainment in the 'frontal cortex', (**d**) cognitive performance in the 'frontal cortex', (**e**) brain shape in cranial neural crest cell ('CNCC'), and (**f**) facial morphology in 'CNCC'.

regulations. For educational attainment, SpecVar revealed 7611 SNP-TG pairs in its SNP-associated regulatory network of 'frontal cortex', and we found that 3693 SNPs (40.1%) of these SNP-TG pairs were also SNPs of eQTL in brain tissues. Among the 2862 SNPs, 788 SNPs (10.4%) had the same TGs with the eGenes in the eQTL database (hypergeometry test, $p = 7.0 \times 10^{-121}$). For cognitive performance, there were 7494 SNP-TG pairs in its SNP-associated regulatory network of 'frontal cortex'. And 3569 of them (47.6%) were also SNPs of eQTL in brain tissues. SpecVar further revealed that 988 SNPs (13.2%) had the same TGs with the eGenes in the eQTL database ($p = 3.2 \times 10^{-224}$). For LDL, there were 556 SNP-TGs pairs in its SNP-associated regulatory network of 'right lobe of liver', and we found that 45 of the SNP-TG pairs (8.1%) were also SNPs of eQTL. Two of the SNPs had the same TGs with the eGenes in the eQTL database ($p = 5.0 \times 10^{-2}$). For total cholesterol, there were 461 SNP-TGs pairs in its SNP associated regulatory network of 'right lobe of liver'. We found that 16 of the SNP-TG pairs (3.5%) were also SNPs of eQTL. There were two of the SNPs that had the same TGs with the eGenes in the eQTL database ($p = 3.0 \times 10^{-2}$).

In summary, we evaluated SpecVar's ability to identify relevant tissues using six well-studied phenotypes as the gold standard by comparison with SAP, SEG, CoCoNet, and RolyPoly. The results showed that SpecVar could identify relevant tissues more accurately and stably. Meanwhile, SpecVar provided detailed regulations to interpret the relevance to tissues and help the fine mapping of GWAS signals.

## SpecVar reveals the association of multiple phenotypes by relevance correlation

SpecVar's ability to accurately and robustly identify relevant tissues enlightens us to define the relevance correlation of two phenotypes by Spearman correlation of their *R* scores (Materials and methods). The relevance correlation may approximate phenotypic correlation since if two phenotypes are correlated, their relevance to human contexts will also be correlated. We used two GWAS datasets with phenotypic correlation computed from individual phenotypic data as the gold standard and compared SpecVar with two other methods SAP and SEG.

The first dataset was GWAS of 78 distances on the human face (*Xiong et al., 2019*). Based on summary statistics, we computed the relevance correlations of 3003 pairs of distances with SpecVar, SAP, and SEG. We compared the relevance correlation to phenotypic correlation from individual phenotypic data and computed the Pearson correlation coefficient (PCC, Materials and methods) to evaluate the performance of these three methods. SpecVar's relevance correlation showed the best performance in approximating phenotypic correlation (*Figure 5a and b*, PCC = 0.522), which outperformed the other two methods: SAP PCC = 0.467 (*Figure 5b*) and SEG PCC = 0.405 (*Figure 5b*). We also evaluated the ability to approximate the phenotypic correlation of highly correlated phenotypes. By setting the threshold of phenotypic correlation to 0.4, we obtained the 363 highly correlated phenotype pairs of facial landmark distances and compared the three methods based on their performance on these 363 pairs of phenotypes. We found that SpecVar also performed best with PCC 0.467, which was the largest among the three methods: SAP PCC = 0.454 and SEG PCC = 0.245 (*Figure 5c*). We also used the mean square error (*Figure 5—figure supplement 1a and b*) and mutual information (*Figure 5—figure supplement 1c and d*) as metrics to evaluate the performance (Materials and methods) and SpecVar was the best among these three methods.

The second GWAS dataset is from UK Biobank. There were 4313 GWAS in UK Biobank, from which we selected 206 high-quality GWAS summary statistics of 12 classes (*Supplementary file 1d*, Materials and methods). We applied SpecVar and the other two methods to obtain the relevance correlations among these 206 phenotypes and used the phenotypic correlations computed from individual data as validation. First, SpecVar performed best in the approximation of phenotypic correlation (PCC = 0.360), followed by SAP (PCC = 0.315) and SEG (PCC = 0.285) (*Figure 5d*). For highly correlated phenotypes, SpecVar's relevance correlation was also closest to phenotypic correlation (*Figure 5e*). SpecVar's outperformance of estimating phenotypic correlation was reproduced by using mean square error (*Figure 5—figure supplement 2a and b*) and mutual information (*Figure 5—figure supplement*

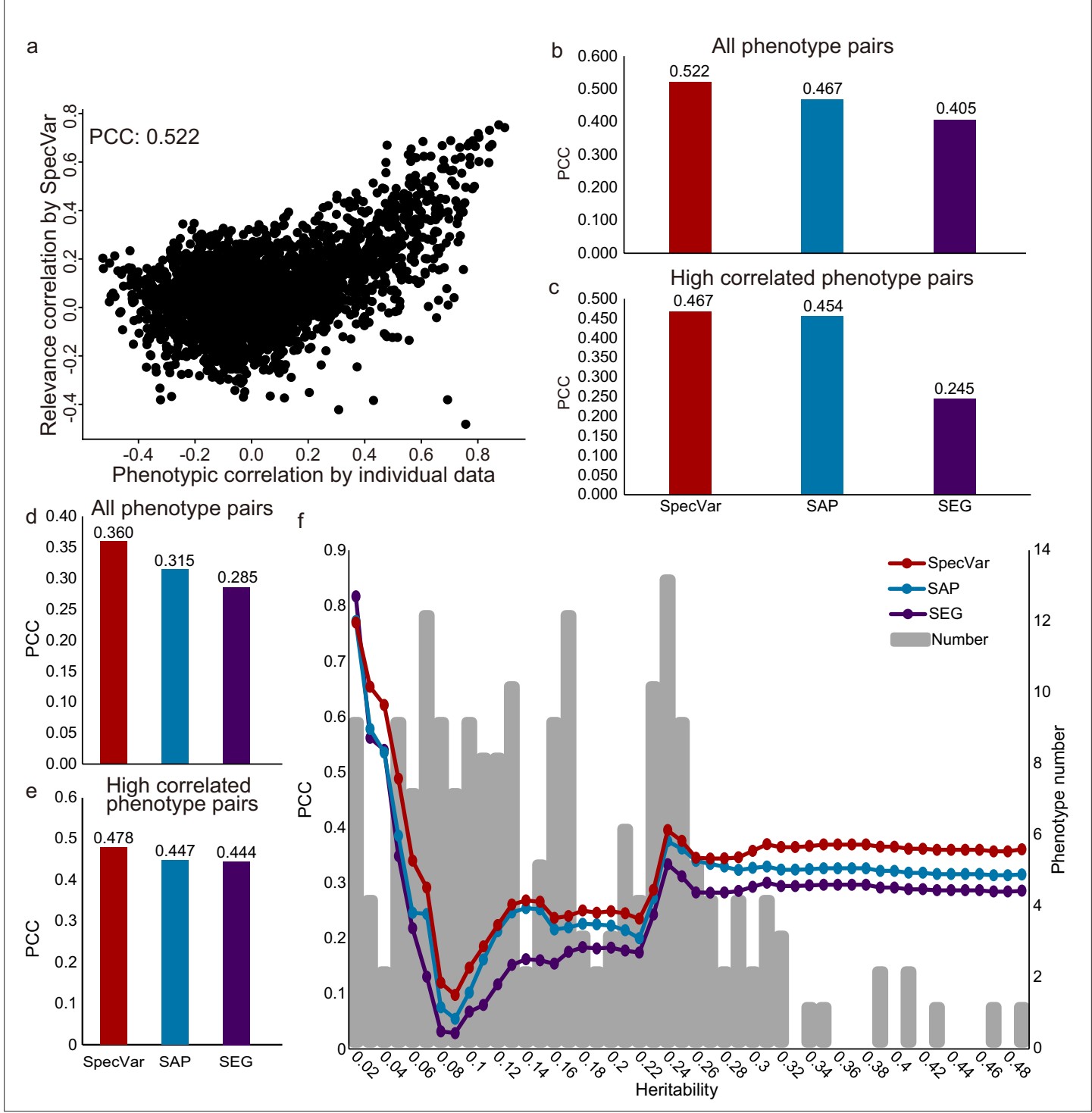

**Figure 5.** SpecVar defines relevance correlation to reveal association of phenotypes. (**a**) The scatter plot of true phenotypic correlation and relevance correlation by SpecVar. Each point means a pair of facial distances. (**b**) For all phenotype pairs of facial distances, the Pearson correlation coefficient (PCC) between phenotypic correlation and relevance correlation of three methods. (**c**) For highly correlated phenotype pairs of facial distances, the PCC between phenotypic correlation and relevance correlation of three methods. (**d**) For all pairs of UKBB phenotypes, the PCC between phenotypic correlation and relevance correlation of three methods. (**e**) For highly correlated pairs of UKBB phenotypes, the PCC between phenotypic correlation and relevance correlation of three methods. (**f**) For UKBB phenotype pairs with 25 different heritability thresholds, the PCC between phenotypic correlation and relevance correlation of four methods.

The online version of this article includes the following figure supplement(s) for figure 5:

*Figure 5 continued on next page*

*Figure 5 continued*

**Figure supplement 1.** Mean square error (MSE) and mutual information (MI) metrics show SpecVar achieve better approximation for phenotypic correlation on facial distance dataset.

**Figure supplement 2.** SpecVar achieve better and more robust approximation for phenotypic correlation on UKBB dataset.

**Figure supplement 3.** SpecVar achieve better approximation for phenotypic correlation than heritability enrichment and p-value on facial distance dataset.

**Figure supplement 4.** Combination of SpecVar's relevance correlation and LDSC-GC's genetic correlation gives a more accurate estimation of phenotypic correlation.

*2c and d*) as metrics. We found that the heritability of these 206 phenotypes was quite variable. For example, 'rose wine intake' had a heritability of $6.5 \times 10^{-3}$ and 'corneal resistance factor right' had a heritability of 0.336. So, we checked whether the heritability would influence the quality of relevance correlation. To do this, we set different thresholds of heritability and obtained a subset of phenotypes for each threshold. Then for the phenotype subset of each heritability threshold, we computed the PCC between relevance correlation and phenotypic correlation. For almost all the thresholds of heritability, SpecVar showed the best performance of PCC and mean square error (*Figure 5f*, *Figure 5— figure supplement 2e*). SpecVar also had the smallest variance regarded heritability among these three methods (*Figure 5—figure supplement 2f and g*). This means that the relevance correlation of SpecVar could estimate phenotypic correlation more accurately and robustly.

SpecVar can interpret the relevance correlation by the common relevant tissues and shared SNP-associated regulations of two phenotypes (*Supplementary file 1e*). For example, 'body mass index' and 'leg fat-free mass (right)' were correlated with a phenotypic correlation of 0.697. SpecVar obtained a relevance correlation of 0.602, while SAP obtained a relevance correlation of 0.342 and SEG gave a relevance correlation of 0.437. SpecVar further revealed that these two phenotypes were correlated because they were both most relevant to the 'frontal cortex' (*Figure 6a*). Body mass index has been reported to be related to frontal cortex development (*Laurent et al., 2020*) and relevant to the reduced and thin frontal cortex (*Islam et al., 2018*; *Shaw et al., 2018*). Obesity and fat accumulation are also revealed to be associated with the frontal cortex (*Gluck et al., 2017*; *Kakoschke et al., 2019*). SpecVar then extracted these two phenotypes' SNP-associated regulatory networks in the 'frontal cortex' and found that their SNP-associated networks were significantly shared. The significant overlapping was observed at SNP, RE, TG, and TF levels: $p = 8.2 \times 10^{-63}$ for SNPs, $p = 1.4 \times 10^{-47}$ for REs, $p = 6.0 \times 10^{-25}$ for TGs, and $p = 8.2 \times 10^{-25}$ for TFs (*Figure 6b*). The shared regulatory network was involved with body weight and obesity. For example, in the brain, *SH2B1* enhances leptin signaling and leptin's antiobesity action, which is associated with the regulation of energy balance, body weight, and glucose metabolism (*Rui, 2014*). We found that one common significant SNP '16:2896143'of these two phenotypes was located in the specific REs of the 'frontal cortex' at the 90k downstream of *SH2B1*. Even though this RE was near the promoter of *NFATC2IP,* SpecVar revealed that it regulated the expression of *SH2B1*, which was supported by a HiChIP loop in brain tissues to associate the SNP-associated RE and promoter of *SH2B1* (*Figure 6c*). These results indicated that one shared SNP was located in the 'frontal cortex'-specific RE and might regulate the expression of *SH2B1* to influence two phenotypes: 'body mass index' and 'leg fat-free mass (right)'.

Through the application of relevance correlation to two datasets with the gold standard of phenotypic correlation, we concluded that SpecVar can use the accurate relevance score to define relevance correlation, which could better estimate phenotypic correlation. SpecVar could further reveal common relevant tissues and shared SNP-associated regulatory networks to interpret correlation of phenotypes.

## Discussion

In this article, we introduce the context-specific regulatory network, which integrate paired gene expression and chromatin accessibility data, to construct context-specific regulatory categories for better interpretation of GWAS data. SpecVar is developed as a tool to interpret genetic variants based on GWAS summary statistics. The key message is that integrating chromatin accessibility and gene expression data into context-specific regulatory networks can provide better regulatory categories for heritability enrichment (*Gazal et al., 2019*). SpecVar is based on the popular model stratified

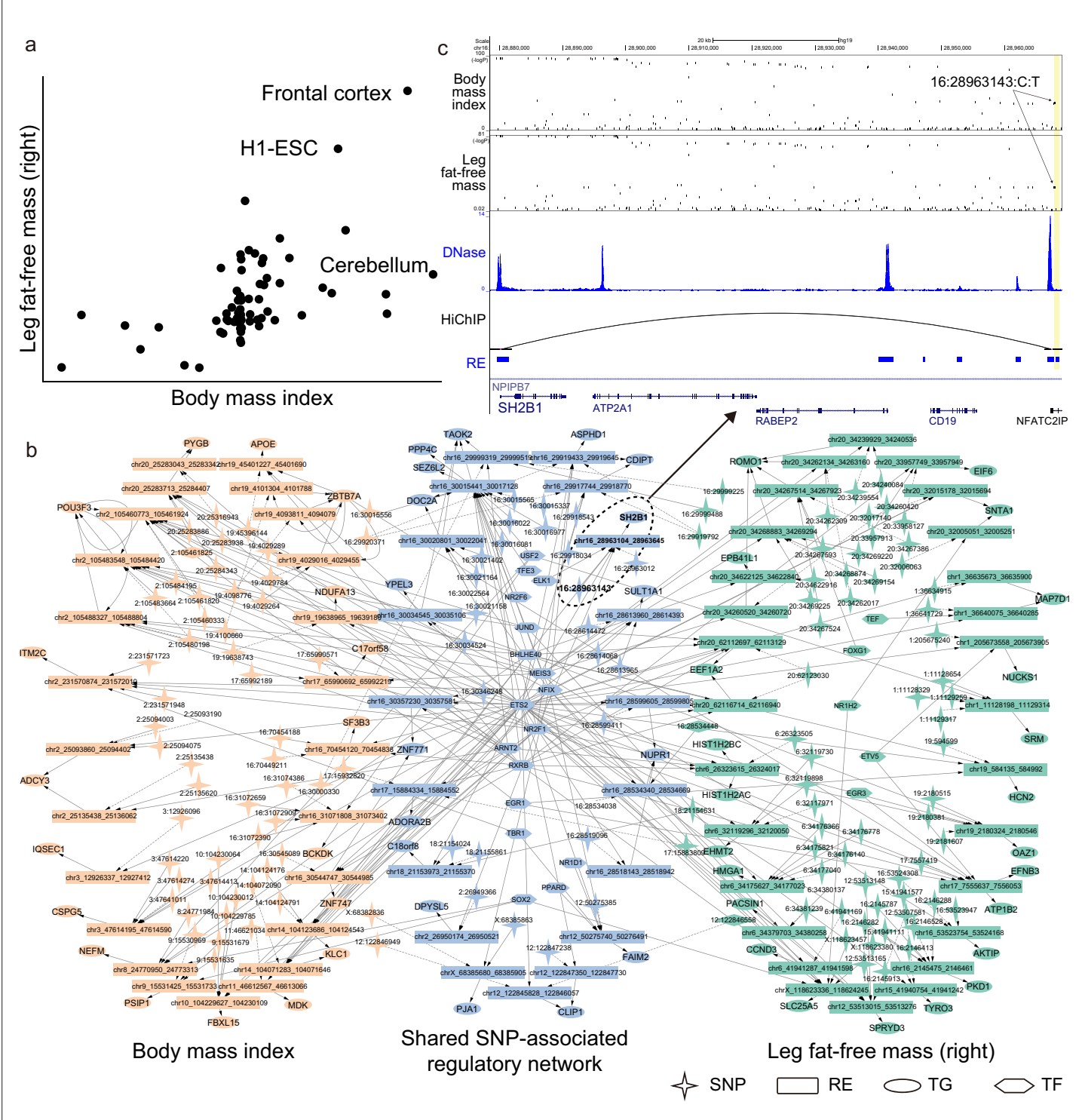

**Figure 6.** SpecVar uses common relevant tissues and shared SNP-associated regulatory network to interpret relevance correlation. (**a**) Scatter plot of *R* scores across 77 human contexts of 'body mass index' and 'leg fat-free mass (right)'. (**b**) The SNP-associated regulatory network of 'body mass index' and 'leg fat-free mass (right)' are significantly shared. (**c**) SNP-associated regulation of *SH2B1*. There is a shared significant SNP of 'body mass index' and 'leg fat-free mass (right)' that is located at the 90k downstream of *SH2B1*, and there is a HiChIP loop linking this locus to the promoter of *SH2B1*. SpecVar prioritizes SNP located in a 'frontal cortex'-specific regulatory elements (RE) as potential causal genetic variant affecting both 'body mass index' and 'leg fat-free mass (right)' through regulation of *SH2B1*.

LDSC (*Finucane et al., 2015*), which includes 52 function categories as the baseline model. In addition, we show that extending the functional categories from noncontext-specific regions to context-specific regions can improve the heritability enrichment, which is consistent with other studies based on gene expression (*Finucane et al., 2018*) and ChIP-seq (*van de Geijn et al., 2020*) data.

Our main contribution in SpecVar is from (1) integrating paired gene expression data and chromatin, which are two types of most easily accessible data, into regulatory networks; (2) highlighting the comparison with other contexts by the specificity of the regulatory network to narrow down the genes and REs, which will help us explain how the phenotype-associated SNPs influence the tissue or cell types in a specific way; and (3) more interpretability than existing methods because SpecVar gives more biological insights in explaining the relevance to tissues by SNP-associated regulatory networks and interpreting phenotype correlation through common relevant tissues and shared SNP-associated regulatory network. To study the contribution of two important parts of SpecVar to the improvement of heritability enrichment: (1) regulatory network by integrating gene expression and chromatin accessibility data, and (2) specificity by comparing with other contexts, we conducted ablation analysis. First, we sorted out the main components of SpecVar (*Figure 2—figure supplement 1c*) and illustrated the relationship and differences of the five methods. Then we compared SpecVar with ARE, SAP, and SEG to show the effect of specificity, gene expression annotations, and chromatin accessibility (*Figure 2—figure supplement 1d*), respectively. We conducted these comparisons for six phenotypes and computed fold change of heritability enrichment to measure the effect to improve heritability enrichment. We found that the annotation from chromatin accessibility contributed most to improve heritability enrichment (*Figure 2—figure supplement 1e*). This meant that the regulatory networks of SpecVar, which integrated chromatin accessibility with gene expression data, played a more important role in improving heritability enrichment. And specificity also improved heritability enrichment by four-fold.

SpecVar outperformed the existing methods in three points. First, SpecVar defined the relevance score based on both heritability enrichment and p-value. Because of the variability in the number of REs in the regulatory categories (*Supplementary file 1f*), using only heritability enrichment or p-value will not give a stable estimation of the relevance of phenotype to tissues. For example, in the experiment of identifying six phenotypes' relevant tissues, heritability enrichment could select the most relevant tissues for LDL and total cholesterol to be the 'right lobe of liver' but failed to get proper tissues for other four phenotypes (*Figure 3—figure supplement 3*). P value could obtain proper tissues for cognitive performance ('frontal cortex') and brain shape (CNCC) but failed to get relevant tissues for LDL, total cholesterol, educational attainment, and facial morphology (*Figure 3—figure supplement 4*). By combining heritability enrichment and p-value into $R$ score, SpecVar could prioritize proper relevant tissues for all the six phenotypes (*Figure 3*). Like the $R$ score-based relevance correlation, we could use the heritability enrichment and p-value to compute relevance correlation (*Figure 5—figure supplement 3a and b*). We found relevance correlation based on heritability enrichment and p-value would give larger MSE (*Figure 5—figure supplement 3c and e*) and lower PCC (*Figure 5—figure supplement 3d and f*) than the $R$ score, which showed that SpecVar's $R$ score can achieve a better approximation of phenotypic correlation. Those comparisons showed that the $R$ score was a good metric to evaluate tissue's relevance to the phenotype. Second, SpecVar's regulatory categories had advantages over the existing functional categories to explain heritability. The context-specific regulatory networks formed regulatory categories that enable better heritability enrichment than other methods (*Figure 2*). The improved heritability enrichment was also observed when we pooled regulatory categories of all the methods together to fit stratified LDSC and re-estimated the heritability enrichment of each context for each method (*Figure 2—figure supplement 2*). The regulatory categories of SpecVar can be used to calculate $R$ scores to identify relevant tissues more accurately than other methods (*Figure 3*). And the $R$ score of SpecVar can also be used to compute relevance correlation to better approximate phenotypic correlation than other methods when we do not have comprehensive phenotype measurement in each individual (*Figure 5*). Third, with the constructed context-specific regulatory network atlas, SpecVar could further interpret the relevant tissue by SNP-associated regulatory networks (*Figure 4*) and interpret relevance correlation by common relevant tissues and shared SNP-associated regulations in relevant tissues (*Figure 6*). These three aspects made SpecVar an interpretable tool for heritability enrichment, identifying relevant tissues, and accessing associations of phenotypes.

Identification of relevant contexts can be conducted at different resolutions. First, in this article, SpecVar identified relevant contexts at the tissue level because most of the 77 contexts were tissues or cell lines. Second, we could identify relevant organs or groups of tissues. For example, our 77 human contexts can be hieriatically clustered into 36 groups. We pooled the REs in the same groups together to form 36 group-level RE sets. Then we defined an RE to be a group-specific RE if it was not overlapped with REs in other groups. This procedure would form 36 sets of group-specific REs. Finally, we fitted the stratified LDSC model with these 36 sets of group-specific regulatory categories and obtained heritability enrichment, p-value, and *R* score for each group by SpecVar. We tested the group-based SpecVar on our six phenotypes. For LDL and total cholesterol, SpecVar identified 'Liver' tissues to be the most relevant tissue (*Figure 3—figure supplement 5a and b*). For educational attainment and cognitive performance, SpecVar revealed 'brain' tissues to be the most relevant tissues (*Figure 3—figure supplement 5c and d*). And for brain shape and facial morphology, SpecVar found 'CNCC' to be the most relevant tissue (*Figure 3—figure supplement 5e and f*). Third, we can identify relevant contexts at the cell-type level since the amount of single-cell multimodal data, such as single-cell RNA-seq and single-cell ATAC-seq data, are increasing in recent years (*Han et al., 2020*). The paired single-cell multi-omics data, which means single-cell data profiled from the same context, enable us to identify cell types and infer regulatory networks for these cell types (*Duren et al., 2018*; *Zeng et al., 2019*). It will be promising to construct an atlas of context-specific regulatory networks at cell-type level and build SpecVar model based on these cell-type-specific regulatory networks. This extension of SpecVar to single-cell level holds the promise to identify more detailed relevant cell types for given phenotypes.

Based on the accurate and highly interpretable relevant tissue identification, the relevance correlation of SpecVar provides us with another perspective of associations between two phenotypes: if two phenotypes are correlated, their relevance to human contexts will also be correlated. This rationale is independent of genetic correlation, which is the proportion of variance that two phenotypes share due to genetic causes and can be estimated with GWAS summary statistics by LDSC-GC (*Bulik-Sullivan et al., 2015a*). When using measured phenotype value correlation as the gold standard of phenotype correlation, we found that SpecVar performed better when the heritability of phenotype was low while LDSC-GC performed better when the heritability was high (*Figure 5—figure supplement 4a and b*). This indicated that the integration of relevance correlation and genetic correlation might give a better estimation of phenotypic correlation. We validated this idea by regressing phenotypic correlation on relevance correlation and genetic correlation in two GWAS datasets. For the phenotypes of facial distances, if we only used relevance correlation to regress phenotypic correlation, the coefficient of determination ($R^2$) was 0.2720; if we only used genetic correlation, the $R^2$ was 0.0002; if we used the linear combination of relevance correlation and genetic correlation to regress phenotypic correlation, the $R^2$ was 0.2765, which was significantly higher than that only with SpecVar (F-test of $R^2$ increase, $p \leq 1.8 \times 10^{-5}$) or only with LDSC-GC ($p \leq 5.3 \times 10^{-213}$); and if we used a product (nonlinear combination) of relevance correlation and genetic correlation, the $R^2$ was much higher: 0.2911 (*Figure 5—figure supplement 4c and d*). And for 206 phenotypes of UK Biobank, if we only used relevance correlation, the $R^2$ was 0.1289; if we only used genetic correlation, the $R^2$ was 0.5614; if we used the linear combination of relevance correlation and genetic correlation to regress phenotypic correlation, the $R^2$ was 0.5927, which was significantly higher than that only with SpecVar ($p \leq 2.2 \times 10^{-16}$) or only with LDSC-GC ($p \leq 2.2 \times 10^{-16}$); and if we used a product of relevance correlation and genetic correlation, the $R^2$ was 0.7375, which was much more improved (*Figure 5—figure supplement 4e and f*). These results showed that relevance correlation and genetic correlation revealed the association of phenotypes in a complementary way.

Our work can be improved in several aspects. The usage of context-specific regulatory networks contributed most to the improvement of SpecVar. But the context-specific regulatory networks can only cover part of the regulatory elements and genetic variants, which are highly essential and representative. Higher-quality and more comprehensive regulatory networks will help obtain better representation. Currently, we built the atlas of regulatory networks of 77 human contexts and only included CNCC in the early developmental stage, which was far from complete. We expect more developmental stages will be included with multi-omics data from ENCODE (*Snyder et al., 2020*) and GTEx (*Consortium, 2020*). On the other hand, SpecVar is based on stratified LDSC, which may not perform well on admixed populations. Thus, considering the effect of mixed ancestries will help

SpecVar handle more circumstances. For example, Luo. et al. have developed cov-LDSC to adjust the LDSC model to be suitable for admixed populations (*Luo et al., 2021*). Building SpecVar model based on cov-LDSC will be promising to perform well on GWAS with mixed populations. Lastly, it will be useful to extend the current approach using a model based on individual whole-genome sequencing data (*Li et al., 2020*).

## Materials and methods

### Regulatory network inference with paired expression and chromatin accessibility data by PECA2

The regulatory networks were inferred by the PECA2 (*Duren et al., 2020*) model with paired expression and chromatin accessibility data. First, we collected paired expression and chromatin accessibility data of 76 human tissue or cell lines from ENCODE and ROADMAP (*Supplementary file 1a*). Then with paired expression and accessibility data of each context, PECA2 calculated two scores. One was the trans-regulatory score. Specifically, PECA2 hypothesized that TF regulated the downstream TG by binding at REs. The trans-regulatory score was calculated by integrating multiple REs bound by a TF to regulate TG to quantify the regulatory strength of this TF on the TG. And PECA2 also considered a prior TF-TG correlation across external public data from ENCODE database. In detail, the trans-regulatory score $TRS_{ij}$ of $i$th TF and $j$th TG was quantified as

$$TRS_{ij} = \left( \sum_k B_{ik} O_k I_{kj} \right) \times 2^{|R_{ij}|} \times \sqrt{TF_i TG_j} \tag{3}$$

Here, $TF_i$ and $TG_j$ are the expressions of the $i$th TF and $j$th TG. $B_{ik}$ is the motif binding strength of $i$th TF on $k$th RE, which was defined as the sum of the binding strength of all the binding sites of $i$th TF on $k$th RE. $O_k$ is the measure of accessibility for $k$th RE. $I_{kj}$ represents the interaction strength between $k$th RE and $j$th TG, which was learned from the PECA model on diverse ENCODE cellular contexts (*Duren et al., 2017*; *Duren et al., 2018*). $R_{ij}$ is the expression correlation of $i$th TF and $j$th TG across diverse ENCODE samples. The significance of the TRS was obtained by a background of randomly selected TF-TG pairs and the threshold of the TRS was decided by controlling the false discovery rate (FDR) at 0.001.

The other one was the cis-regulatory score to measure the regulatory strength of RE on a TG. The cis-regulatory score $CRS_{kj}$ of $k$th RE on $j$th TG was quantified as

$$CRS_{kj} = \left( \sum_i B_{ik} TRS_{ij} \right) \times I_{kj} \times O_k \tag{4}$$

We approximated the distribution of $\log_2 \left( 1 + CRS_{kj} \right)$ by a normal distribution and predicted RE-TG associations by selecting the RE-TG pairs that had p-value≤0.05.

The output of PECA2 was a regulatory network with TFs, REs, and TGs as nodes and the regulations among them as edges. This procedure was applied to 76 human contexts with paired expression and chromatin accessibility data and obtained 76 regulatory networks. We noted that the regulatory network of the early development stage CNCC was reconstructed recently (*Feng et al., 2021*), and we included the regulatory network of CNCC to form our regulatory network atlas of 77 human contexts.

### Construction of context-specific regulatory network atlas and regulatory categories

The context-specific regulatory network was obtained based on the specificity of REs. In detail, we had 77 regulatory networks, and each regulatory network had a set of REs $RE_i, 1 \leq i \leq 77$. Firstly, we hierarchically clustered 77 contexts' regulatory networks into 36 groups by trans-regulatory scores (*Supplementary file 1a*). Then for a given context, a RE was defined as a context-specific RE if it was not overlapped with REs of other contexts. Formally, the context-specific RE set of $i$th context $C_i$ was defined as

$$C_i = \left\{ RE_{ik} \in RE_i | RE_{ik} \notin RE_j, j \neq i \right\} \tag{5}$$

Here, $RE_{ik} \notin RE_j$ means $RE_{ik}$ was not overlapped with any REs in $RE_j$:

$$RE_{ik} \notin RE_j \iff RE_{ik} \text{ is not overlapped with any } RE_{jl} \text{ in } RE_j \qquad (6)$$

And we defined 'overlapped' (1) for REs from contexts of the different groups, two REs were overlapped if their overlapping base ratio was over 50%; (2) for REs from contexts of the same group, two REs were overlapped if their overlapping base ratio was over 60%. The reason we used different 'overlapped' criteria for REs from the same group and different groups was to retain group-specific REs. For example, we had five cell types for the brain tissues: 'Ammon's horn', 'caudate nucleus', 'cerebellum', 'frontal cortex', and 'putamen'. If we defined RE's specificity with stringent conditions among these five brain cell types, many REs of brain lineage would be lost.

Finally, the context-specific regulatory network was formed by specific REs and their directly linked upstream TFs and downstream TGs. And the context-specific RE sets $C_i, 1 \leq i \leq 77$ gave the regulatory categories of SpecVar.

## Heritability enrichment and *R* score of GWAS summary statistics by SpecVar

SpecVar used stratified LDSC (*Finucane et al., 2015*) to compute partitioned heritability enrichment. Under the linear additive model, stratified LDSC models the causal SNP effect on phenotype as drawn from a distribution with mean zero and variance

$$Var\left(\beta_j\right) = \sum_i \tau_i 1_{\{j \in C_i\}} \qquad (7)$$

And with the assumption that the LD of a category that is enriched for heritability will increase the $\chi^2$ statistic of an SNP more than the LD of a category that does not contribute to heritability, the expected $\chi^2$ statistic is modeled as follows:

$$E\left(\chi_j^2\right) = N \sum_{C_i} \tau_i l\left(j, i\right) + Na + 1 \qquad (8)$$

where $N$ is the sample size; $C_i$ denotes the regulatory category formed by the $i$th context-specific regulatory network; $\chi_j^2$ is the marginal association of SNP $j$ from GWAS summary statistics; $r_{jk}^2$ is the LD score of SNP $j$ in the $i$th category, where $r_{jk}^2$ was the genotype correlation between SNP $j$ and SNP $k$; $a$ measures the contribution of confounding biases; and $\tau_i$ represents heritability enrichment of SNPs in $C_i$. Stratified LDSC estimates standard error with a block jackknife and uses the standard error to calculate the p-value $p_i$ for the heritability enrichment $\tau_i$ (*Finucane et al., 2015*).

To make a trade-off between the heritability enrichment score and p-value resulting from a hypothesis test, we combined heritability enrichment and statistical significance (p-value) to define the relevance score ($R_i$) of this phenotype to $i$th context as follows:

$$R_i = \tau_i \cdot \left(-\log p_i\right) \qquad (9)$$

The relevance score (*R* score) offered a new robust means to rank and select relevant tissue for a given phenotype (*Xiao et al., 2014*; *Figure 3—figure supplements 3 and 4*, and *Figure 5—figure supplement 3*).

## Significance testing of *R* score and identification of relevant contexts

We used block jackknife to estimate the standard error, p-value, and FDR for the *R* score. Specifically, we computed the *R* score $R_i$ of a phenotype to $i$th context. Then we estimated the standard error, p value, and FDR of $R_i$ following the procedures:

1. For the $i$th context, we divided its specific RE into 100 folds.
2. One subsample was generated by removing one of the 100 folds, and we generated 100 subsamples of $i$th context.
3. These 100 subsamples of specific REs would form new regulatory categories for fitting stratified LDSC. For each subsample, we obtained heritability enrichment, p-value, and *R* score by SpecVar.

4. With the 100 background $R$ scores of the 100 subsamples, we could estimate the standard error ($SD_i$) of $R_i$ for $i$th context.
5. We computed the z scores of $i$th context: $Z_i = R_i/SD_i \sim N(0, 1)$, and estimated the p-value and FDR q-value.

The $R$ scores and their FDR q-values could be used to select relevant tissues. We used $R$ score ≥100 and FDR ≤0.01 as the threshold to pick up relevant tissues for a phenotype. For six well-studied phenotypes in this article, their relevant tissues, $R$ scores, and FDR are summarized in *Table 1*.

## Four alternative methods to construct representations of GWAS summary statistics

Based on expression and chromatin accessibility data, there were four alternative methods for constructing regulatory categories: AAP, SAP, SEG, and ARE.

The AAP method used all the chromatin-accessible peaks of each context to form a genome functional category, which was used for partitioned heritability enrichment analysis. The SAP method used the same rules of SpecVar above to obtain specifically accessible peaks of each context, and the context-specific peaks sets of $M$ contexts formed functional categories of SAP. The SEG method was constructed by following the procedure in *Finucane et al., 2018*. First, the $t$-statistics for differential expression of each gene in each of the $M$ contexts were calculated. Then for each context, the top 10% genes ranked by $t$-statistic were selected, and the 100 kb windows around those top 10% genes were used to form a functional category. For the ARE method, we obtained all REs in the regulatory network of a context to be a functional category, and the RE sets of $M$ contexts formed regulatory categories of ARE.

We could conclude the difference and relationship between the five methods (*Figure 2—figure supplement 1c*): SEG was the combination of gene expression and specificity; AAP was only from chromatin accessibility; SAP was the combination of chromatin accessibility and specificity; ARE integrated gene expression and chromatin accessibility; and SpecVar considered gene expression, chromatin accessibility, and specificity. *Supplementary file 1f* shows the number and size of genomic regions of each method and the overlapping among these methods.

After obtaining functional categories with these four alternate methods, we could also use stratified LDSC to obtain heritability enrichment and define the $R$ score representation of GWAS summary statistics with *Equations 8 and 9*. We called them AAP, SAP, SEG, and ARE, respectively. We compared these four alternate methods with SpecVar.

## Relevance correlation analysis of SpecVar

SpecVar defined relevance correlation based on $R$ scores. The $R$ scores to $M$ contexts could be aggregated into a context-specific vector representation of GWAS summary statistics:

$$R = (R_1, R_2, \cdots, R_M) \tag{10}$$

For two phenotypes, such as phenotype $p$ and phenotype $q$, we obtained their $R$ score representations:

$$R^p = (R_1^p, R_2^p, \cdots, R_M^p)$$
$$R^q = (R_1^q, R_2^q, \cdots, R_M^q) \tag{11}$$

Then the Spearman correlation of their $R$ score representation was used to define the relevance correlation:

$$\rho_g = \rho(R^p, R^q) = 1 - \frac{6 \sum_i \left[r(R_i^p) - r(R_i^q)\right]^2}{M * (M^2 - 1)} \tag{12}$$

Here, $r(R_i^p)$ and $r(R_i^q)$ are the ranks of $i$th context by the $R$ score for the two phenotypes.

For two other specificity-based regulatory categories SAP and SEG, we also used their functional categories to compute heritability enrichment and p-value and defined the $R$ score with *Equations 7–9*. The $R$ scores of SAP and SEG were used to compute relevance correlation.

## Evaluation of relevant tissue identification and relevance correlation

To evaluate the performance of the SpecVar and other methods, we used different datasets as the gold standard.

For the application to identify relevant tissues, we used six well-studied phenotypes that we had knowledge of the relevant tissues: two lipid phenotypes (LDL and total cholesterol) were relevant to the liver; two human intelligential phenotypes (educational attainment and cognitive performance) were relevant to the brain; two craniofacial bone phenotypes (facial morphology and brain shape) were relevant to CNCC. We used different methods to identify relevant tissues of these six phenotypes and checked whether they obtained the proper relevant tissues.

For relevance correlation, we used the phenotypic correlation computed with individual phenotypic data as the gold standard. First, we computed the PCC between relevance correlation and phenotypic correlation:

$$PCC = \frac{\sum_{i,j \in P} (p_{ij} - \bar{p})(p'_{ij} - \bar{p}')}{\sqrt{\sum_{i,j \in P} (p_{ij} - \bar{p})^2 \sum_{i,j \in P} (p'_{ij} - \bar{p}')^2}}$$ (13)

Here, $P$ is the set of phenotypes, and $N$ is the number of phenotype pairs; $p_{ij}$ is the phenotypic correlation computed with individual phenotypic data, and $p'_{ij}$ is the relevance correlation; $\bar{p}$ is the average of $p_{ij}$, and $\bar{p}'$ is the average of $p'_{ij}$. A larger PCC indicated better performance in approximating phenotypic correlation.

We also used the mean square error (MSE) between relevance correlation and phenotypic correlation:

$$MSE = \frac{\sum_{i,j \in P} (p_{ij} - p'_{ij})^2}{N}$$ (14)

A smaller MSE indicated better performance in approximating phenotypic correlation.

The last metric we used to evaluate the performance of approximating phenotypic correlation was mutual information, which was a nonlinear metric. We used the 'k-nearest neighbor' stratagem proposed by Kraskov et al. to estimate the mutual information of two vectors (*Kraskov et al., 2004*).

## Extracting SNP-associated regulatory subnetworks in relevant tissues

Given a phenotype's GWAS summary statistics and a context, SpecVar identified SNPs-associated regulatory subnetwork by considering the following two factors: (1) the cis-regulatory score of SNP-associated RE should be large enough to indicate its importance in the regulatory network; (2) the risk signal of SNPs (i.e., p-value) on or near this RE should be large to indicate its association with phenotype. We combined these two factors to define the association score ($A$ score) of SNP-associated REs.

First, the regulatory strength of $k$th RE was measured by the maximum cis-regulatory score of this RE. Formally,

$$C_k = \max_j CRS_{kj}$$ (15)

Here, $CRS_{kj}$ is the cis-regulatory score of $k$th RE on $j$th TG. For the $k$th RE, the larger $C_k$ was, the more important role this RE played in the regulatory network. Second, the risk score of GWAS $S_k$ for $k$th RE was defined as the average of the -log(p-value) of SNPs located on or near this RE, which were downweighted by their LD scores and distances to RE:

$$S_k = \frac{1}{|P_k|} \sum_{l \in P_k} -\omega_l \cdot \log(p_l) \cdot e^{-\frac{d_{lk}}{d_0}}$$ (16)

Here, $P_k$ is the set of SNPs whose distances were less than 50 kb to the $k$th RE, and $|P_k|$ is the total number of this SNP set; $\omega_l$ (the reciprocal of LD score, downloaded at https://data.broadinstitute.org/alkesgroup/LDSCORE/) is the weight of the $l$th SNP; $p_l$ is the p-value of the $l$th SNP in summary statistics; $d_{lk}$ is the base pair distance of the $l$th SNP to $k$th RE and $d_0$ is a constant, which was set to 5000 as default. For the $k$th RE, a larger value of $S_k$ indicated a stronger association with the given phenotype.

Finally, we obtained the association score ($A$ score) of $k$th RE by combining these two factors:

$$A_k = \sqrt{C_k * S_k} \tag{17}$$

Every RE in the context-specific regulatory network was qualified by the $A$ score. We used the GWAS of six phenotypes to analyze the distribution of $A$ scores and found that the $A$ scores followed a Gaussian distribution (*Figure 4—figure supplement 1*). So, we hypothesized the distribution of $A$ scores was Gaussian distribution and selected the REs associated with the given phenotype by $A$ scores' FDR threshold of 0.05. The prioritized REs, as well as their directly linked upstream TFs, downstream TGs, and the associated SNPs, formed the SNP-associated regulatory subnetwork.

## GWAS summary statistics of UK Biobank

The GWAS summary statistics of UK Biobank were downloaded at http://www.nealelab.is/uk-biobank. There were 4176 phenotypes and 11,372 GWAS summary statistics. We selected 206 GWAS summary statistics (*Supplementary file 1d*) based on the following conditions.

1. Excluding sex-specific and 'raw'-type GWAS.
2. Sample size condition: $N \geq 50,000$ and $N_{control}$, $N_{case} \geq 10,000$ for binary and categorical phenotypes.
3. Significant SNP number condition: the number of SNPs that pass the threshold of $5 \times 10^{-8}$ was not less than 500.
4. Manually curation: removing duplicated phenotypes, 'job', 'parent', and 'sibling'-associated phenotypes.

## Acknowledgements

We acknowledge funding from the National Key Research and Development Program of China (2022YFA1004800 and 2020YFA0712402), Strategic Priority Research Program of the Chinese Academy of Sciences (XDPB17), CAS Project for Young Scientists in Basic Research, Grant No. R, and the National Natural Science Foundation of China (grants 12025107, 11871463, and 11688101).

## Additional information

### Funding

| Funder | Grant reference number | Author |
|---|---|---|
| National Key Research and Development Program of China | 2022YFA1004800 2020YFA0712402 | Yong Wang |
| Strategic Priority Research Program of the Chinese Academy of Science | XDPB17 | Yong Wang |
| CAS Young Scientists in Basic esearch | YSBR-077 | Yong Wang |
| National Natural Science Foundation of China | 12025107 | Yong Wang |
| National Natural Science Foundation of China | 11871463 | Yong Wang |
| National Natural Science Foundation of China | 11688101 | Yong Wang |

The funders had no role in study design, data collection and interpretation, or the decision to submit the work for publication.

### Author contributions

Zhanying Feng, Conceptualization, Resources, Data curation, Software, Formal analysis, Validation, Investigation, Visualization, Methodology, Writing - original draft, Writing – review and editing; Zhana Duren, Resources, Formal analysis, Writing – review and editing; Jingxue Xin, Yaoxi He, Bing Su, Formal

analysis, Writing – review and editing; Qiuyue Yuan, Formal analysis, Visualization, Writing – review and editing; Wing Hung Wong, Formal analysis, Supervision, Methodology, Project administration, Writing – review and editing; Yong Wang, Conceptualization, Software, Formal analysis, Supervision, Funding acquisition, Validation, Methodology, Project administration, Writing – review and editing

**Author ORCIDs**
Zhanying Feng ![ORCID] http://orcid.org/0000-0002-5727-3929
Yong Wang ![ORCID] http://orcid.org/0000-0003-0695-5273

**Decision letter and Author response**
Decision letter https://doi.org/10.7554/eLife.82535.sa1
Author response https://doi.org/10.7554/eLife.82535.sa2

## Additional files

**Supplementary files**
• Supplementary file 1. Supplementary tables.
 (a) Data source, transcription factor (TF) number, target gene (TG) number, regulatory element (RE) number, RE number per TG, and group information of paired RNA-seq and ATAC-seq for 77 regulatory networks. (b) SpecVar-based heritability enrichment, and its standard error, p-value, and q-value. (c) $R$ scores of six phenotypes in 77 human contexts. (d) 206 phenotypes selected from UKBB and their top relevant contexts. (e) Relevance correlation and the top three common relevant tcontexts of UKBB phenotypes. (f) Number of regions of five genome partition methods (Spec, ARE, SAP, AAP, SEG). ARE, all regulatory elements; AAP, all accessible peaks; SAP, specifically accessible peaks; SEG, specifically expressed gene.

• MDAR checklist

**Data availability**
Codes and regulatory network resources are available at https://github.com/AMSSwanglab/SpecVar, (copy archived at swh:1:rev:cf27438d3f8245c34c357ec5f077528e6befe829, *Feng et al., 2022a*). Expression and chromatin accessibility data were summarized in Supplementary file 1a. GWAS data used: GWAS summary statistics of LDL and TC were downloaded at http://csg.sph.umich.edu/willer/public/lipids2013/; GWAS summary statistics of EA (GCST006442), CP (GCST006572), BrainShape (GCST90012880-GCST90013164), and Face (GCST009464) were downloaded at GWAS catalog https://www.ebi.ac.uk/gwas/summary-statistics; GWAS summary statistics of UK-Biobank were downloaded at http://www.nealelab.is/uk-biobank. The LDSC genetic correlation and phenotypic correlation computed from individual phenotypic data were downloaded at https://ukbb-rg.hail.is/.

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
