## [Editor Report]

In this article, the authors develop a method to identify potentially causal tissues and cell types for complex diseases by performing heritability enrichment estimation using information from gene regulatory networks. This article is of significant interest to geneticists and biologists interested in unraveling the molecular basis of disease. The key claims of the article are well supported by the data. The work has the potential to inform our understanding of the genetics of complex diseases.

---

## [Decision Letter]

**Decision letter after peer review:**

Thank you for submitting your article "Heritability enrichment in context-specific regulatory networks improves phenotype-relevant tissue identification" for consideration by *eLife*. Your article has been reviewed by 3 peer reviewers, and the evaluation has been overseen by a Reviewing Editor and David James as the Senior Editor. The reviewers have opted to remain anonymous.

Essential revisions:

1. Please improve the description of the advance of the approach relative to other publications. How does this combination of an existing network method and an existing heritability enrichment method constitute substantial progress in the field?

2. Reporting of SpecVar-based heritability enrichment point estimates should be accompanied by appropriate measures of statistical significance (p-values, FDR-adjusted q-values, confidence intervals, etc.).

3. The comparisons between SpecVar and other approaches (AAP, ARE, SAP, SEG) are not sufficiently rigorous.

*Reviewer #1 (Recommendations for the authors):*

The approach taken in the study is conceptually interesting, however, it's not clear how novel the approach is. Additionally, there are several issues related to reporting results that need to be addressed. In particular, the authors do not report the statistical significance of enrichments using the new annotation aggregation method. Therefore, it is not possible to evaluate whether the higher overall heritability enrichment of the aggregate annotations they report compared to existing annotation approaches is meaningful.

1. Describe the novelty of the approach relative to other publications. How does this combination of an existing network method and an existing heritability enrichment method warrant a substantial advance in the field?

2. Reporting of SpecVar-based heritability enrichment point estimates should be accompanied by appropriate measures of statistical significance (p-values, FDR-adjusted q-values, confidence intervals, etc.).

3. The comparisons between SpecVar and other approaches (AAP, ARE, SAP, SEG) are not sufficiently rigorous.

a. First, please provide additional context. What is the amount of the genome covered by the regions defined by each approach? What is the overlap between these regions? For example, Table S4 shows SpecVar sets typically include substantially fewer regions than SAP. Are SpecVar regions primarily a subset of SAP regions or do they contain distinct/non-overlapping territory in the genome?

b. Second, please provide a way to evaluate whether differences in enrichment between the methods are statistically meaningful.

c. Fitting a joint LDSC model with multiple annotation sets would provide a more rigorous assessment of the performance of SpecVar relative to the other approaches.

4. Y-axes for bar plots in Figure 5 should start at 0. The floating y-axis origin is highly misleading.

5. The approach to defining context-specific REs based on overlap (>50% of bases overlapping for cross-group comparisons, and >60% of bases overlapping for within-group comparisons) seems quite stringent and likely excludes a lot of lineage-specific REs. Calculating enrichments for "group-specific" regulatory networks, where REs are considered group-specific if they do not overlap any of the other 35 groups resulting from the hierarchical clustering (ignoring whether an RE overlaps REs from other contexts within the same group), might provide more biologically relevant region sets.

*Reviewer #2 (Recommendations for the authors):*

– The authors only show the performance of the SpecVar method on 6 traits, however, it is not clear whether these are representative of all of the traits in UKBiobank. For traits with fewer SNP associations and lower sample numbers it appears that LDSC-SAP produces much high trait relevance scores, however, the authors use different tissues in each method so it may not be a fair comparison.

– The authors use the Pearson correlation coefficient to evaluate the performance of the SpecVar method, however, they should also consider other nonlinear metrics.

– The authors should consider a comparison with other regulatory network-based heritability methods such as CoCoNet, which is based on co-regulated genes. They should also compare to other non-network-based methods.

– The highlighted example SNP-associated network for the FOXC2 variants is interesting, however, the authors should demonstrate whether there are chromatin interactions (HiC or HiChIP) in brain tissues linking these variants in ATAC-seq peaks to the FOXC2 promoter. It would also be helpful to highlight another example using distinct UKBB phenotype and tissue datasets.

– It will be more interesting to adapt this to paired multimodal single-cell data if possible or expand on this in the Discussion.

– Lastly, LDSC-based methods may not perform well on admixed populations, thus some discussion on how this could be adapted using covariate-adjusted approaches (e.g. cov-LDSC) would be helpful.

*Reviewer #3 (Recommendations for the authors):*

The study would benefit from investigating the following questions:

Which part of the new annotation drives the extreme heritability enrichment from SpecVar, compared to other LDSC-based methods?

How to use the R score for a formal hypothesis test, rather than subjectively picking a few top-ranked tissues for the trait? How to evaluate the false positive rate for the test?

How to explain the inconsistent findings with previous studies, e.g., the association of the liver to LDL?

Are the GWAS signal in context-specific RE colocalised with context-specific eQTL signals?

How to link relevance score correlation across tissues to shared heritability between traits?

---

## [Author Response]

Essential revisions:1. Please improve the description of the advance of the approach relative to other publications. How does this combination of an existing network method and an existing heritability enrichment method constitute substantial progress in the field?

We have followed the editors’ and reviewers’ suggestions to add more description about our novelty. We further conducted the ablation analysis to demonstrate the complementary contribution of two components (regulatory network and specificity) of SpecVar. The ablation analysis showed integrating paired gene expression and chromatin accessibility data into regulatory network played an essential role in improving heritability enrichment for SpecVar (Figure 2-figure supplement 1). Together with additional description and discussion, we highlighted SpecVar’s novelty in the integration of multi-omics data, specificity, and interpretability. We thank the editor and reviewers for their kind suggestions, which help us to greatly clarify the novelty and contribution of our work.

2. Reporting of SpecVar-based heritability enrichment point estimates should be accompanied by appropriate measures of statistical significance (p-values, FDR-adjusted q-values, confidence intervals, etc.).

We thank the reviewers and editor for the constructive suggestion for improving our paper. In the revision, we provided more detailed information on heritability enrichment, including p-values, FDR-adjusted q-values, and confidence. We also improved the statistical rigorousness of SpecVar by block Jackknife stratagem to estimate standard error, p-value, and q value of R scores (Figure 3).

3. The comparisons between SpecVar and other approaches (AAP, ARE, SAP, SEG) are not sufficiently rigorous.

According to the suggestions, we performed a more comprehensive comparison with other methods. (1). We conducted Welch's t-test and showed that the differences between SpecVar and other methods were statistically significant (Figure 2). (2). We fitted a joint LDSC model with multiple annotation sets of all methods and provided a more rigorous assessment of SpecVar’s performance to estimate heritability enrichment (Figure 2-figure supplement 2) and identify relevant tissues (Author response image 1) relative to the other approaches. (3). We further compared with relevant tissue identification methods that were not based on stratified LDSC. We chose a network-based method CoCoNet (Figure 3-figure supplement 1) and a non-network-based method RolyPoly (Figure 3-figure supplement 2). The results also showed SpecVar’s outperformance over other methods. (4). We discussed different levels of relevant tissues: we built tissue groups based SpecVar to show identification of relevant tissue groups or organs (Figure 3-figure supplement 5); our current SpecVar revealed relevant contexts were tissues or cell lines; single-cell-based methods would help identify relevant cell types. (5). In evaluating the performance to approximate phenotypic correlation, we used not only the linear metrics (PCC, MSE) but also the non-linear metric (mutual information) to show SpecVar’s relevance correlation was better to estimate phenotype correlation (Figure 5-figure supplement 1, 2).

**Author response image 1. sa2fig1:** The top 5 relevant tissues of SpecVar, SAP, and SEG ranked by the relevant score estimated by “pooled genome partition” for (a) LDL, (b) total cholesterol, (c) educational attainment, (d) cognitive performance, (e) brain shape, and (f) facial morphology.

Reviewer #1 (Recommendations for the authors):The approach taken in the study is conceptually interesting, however, it's not clear how novel the approach is. Additionally, there are several issues related to reporting results that need to be addressed. In particular, the authors do not report the statistical significance of enrichments using the new annotation aggregation method. Therefore, it is not possible to evaluate whether the higher overall heritability enrichment of the aggregate annotations they report compared to existing annotation approaches is meaningful.

We thank the reviewers for the interest in SpecVar. We have followed your suggestion to highlight the novelty and conducted an ablation analysis to understand the contribution from two components of SpecVar (multi-omics data integration into regulatory network and specificity). We added the necessary standard error, p-value, and q value to assess the statistical significance of heritability enrichments. We performed Welch's t-test and fitted stratified LDSC with pooled annotation of all methods to show the improved heritability enrichment was statistically meaningful.

1. Describe the novelty of the approach relative to other publications. How does this combination of an existing network method and an existing heritability enrichment method warrant a substantial advance in the field?

We are sorry that we did not make the novelty of our approach clear in the manuscript. As summarized by the reviewer, our method aims to combine the regulatory network with existing heritability enrichment methods to improve heritability enrichment and help identify relevant tissues. The novelty of our approach over existing stratified LDSC and LDSC-SEG lies in three aspects:

1) Our method integrates paired gene expression data and chromatin, which are two types of most easily accessible data now, into the regulatory network. Regulatory networks are involved with sets of macromolecules, mostly proteins and RNAs, that interact to control the level of expression of various genes in the genome. The gene expression and chromatin accessibility in the regulatory network exert constraints on each other to find the truly active genes and regulatory elements for certain tissues or cell types, which give more accurate biological molecules for interpreting genetic variants. In our previous studies, the regulatory networks have been successfully used to identify the master regulators in stem cell differentiation (Li et al., 2019) and to interpret conserved regions for the non-model organisms (Xin et al., 2020). Regulatory networks have also been used to interpret the non-coding genetic variants in certain tissues or cell types (Feng et al., 2021; Ma et al., 2022; Zhu et al., 2021).

2) Our method highlights the comparison to other contexts by specificity. There are some previous findings that the phenotype-associated SNPs often function in a tissue- or cell-type-specific manner (Westra and Franke, 2014). Considering the specificity of the regulatory network helps us to narrow down the genes and REs to be context-specific genes and REs, which will help us explain how the phenotype-associated SNPs influence the tissue or cell types in a specific way.

3) Our method is more interpretable than existing methods. Through context-specific regulatory networks, SpecVar can improve heritability enrichment, achieve accurate relevance measurement to tissues, and reveal correlated phenotypes by relevance correlation. Besides, SpecVar gives more biological insights in explaining the relevance to tissues by SNP-associated regulatory networks and interpreting phenotype correlation through common relevant tissues and shared SNP-associated regulatory networks.

Except for the conceptional description, we conducted additional ablation analysis to study the contribution of two important components of SpecVar to the improvement of heritability enrichment: regulatory network by integrating gene expression and chromatin accessibility data, and specificity by comparing with other contexts. First, we used (Figure 2—figure supplement 1C) to illustrate the relationship and differences of the five methods: SEG was the combination of gene expression and specificity; AAP was only from chromatin accessibility; SAP was the combination of chromatin accessibility and specificity; ARE integrated gene expression and chromatin accessibility; SpecVar considered gene expression, chromatin accessibility, and specificity. Then we compared SpecVar to ARE to show the effect of specificity, compared SpecVar to SAP to show the effect of gene expression annotations, and compared SpecVar to SEG to show the effect of chromatin accessibility (Figure 2—figure supplement 1d). We conducted these comparisons on six phenotypes and computed fold change of different methods’ heritability enrichment to measure the effect. We found that the annotation from chromatin accessibility give the largest effect to improve heritability (Figure 2—figure supplement 1e). This meant that the regulatory networks of SpecVar, which integrated chromatin accessibility with gene expression data, played a more important role in heritability enrichment. And specificity can also improve heritability enrichment.

We thank the reviewer’s suggestion to highlight the novelty of our method and we have added the description of our novelty and the additional ablation analysis to our revised manuscript.

Excerpt from Manuscript: (Page 3) …… Compared to the existing methods, SpecVar shows novelty in three aspects: (1) SpecVar integrates paired gene expression data and chromatin, which are two types of easily accessible data with rich information, into regulatory networks. The gene expression and chromatin accessibility in the regulatory network are complementary to each other to reveal the high quality active regulatory elements and genes for certain tissues or cell types to interpret genetic variants; (2) SpecVar highlights the comparison to other contexts by specificity to narrow down the regulatory molecules; (3) SpecVar is more interpretable because it can explain the relevance to tissues by SNP-associated regulatory networks and interpret phenotype correlation through common relevant tissues and shared SNP-associated regulatory network ……

(Page 6) …… To explore the heritability enrichment improvement of SpecVar, we conducted ablation analysis to study the contribution of two important parts of SpecVar: (i) regulatory network by integrating gene expression and chromatin accessibility data, and (ii) specificity by comparing with other contexts. Figure 2—figure supplement 1c shows the relationship and difference of the five methods: SEG is the combination of gene expression and specificity; AAP is only from chromatin accessibility; SAP is the combination of chromatin accessibility and specificity; ARE integrates gene expression and chromatin accessibility; and SpecVar considers integration of gene expression and chromatin accessibility, and specificity. To show the effect of the regulatory network in heritability enrichment, we compared SpecVar with SAP and showed the effect of gene expression data. We compared SpecVar with SEG and showed the effect of chromatin accessibility data. We compared SpecVar with ARE and showed the contribution of specificity (Figure 2—figure supplement 1d). To quantify the effect of each component, we computed the fold change of different methods’ heritability enrichments. We found chromatin accessibility, which was part of the regulatory network, showed the highest effect in improving heritability enrichments for all six phenotypes. This is consistent with the fact that most genetic variants are located in the non-coding regulatory regions (Claussnitzer et al., 2015; Kumar et al., 2012; Smemo et al., 2014) and accessibility gives the direct functional evidence for genetic variants. The specificity in SpecVar also contributed at least 4-fold improvement for heritability enrichment (Figure 2—figure supplement 1e) ……

2. Reporting of SpecVar-based heritability enrichment point estimates should be accompanied by appropriate measures of statistical significance (p-values, FDR-adjusted q-values, confidence intervals, etc.).

We have followed your suggestions and added confidence intervals (standard error) of heritability enrichment as error bars into figures of our manuscript (Figure 2). In addition, we provided detailed standard error, p-value, and q-value in Supplementary file 1b. The standard error, p-value, and q-value were estimated by block jackknife.

3. The comparisons between SpecVar and other approaches (AAP, ARE, SAP, SEG) are not sufficiently rigorous.a. First, please provide additional context. What is the amount of the genome covered by the regions defined by each approach? What is the overlap between these regions? For example, Table S4 shows SpecVar sets typically include substantially fewer regions than SAP. Are SpecVar regions primarily a subset of SAP regions or do they contain distinct/non-overlapping territory in the genome?

We have followed your suggestions and provided additional information about the genomic region sets of five methods in Supplementary file 1f. For each context, we presented the amount of genome covered by each method. Since genomic regions of SpecVar were a subset of ARE and genomic regions of SAP were a subset of AAP, we only provided the overlapping among SpecVar, SAP, and SEG for each context. These three methods cover distinct territories in the genome and have overlapping.

Excerpt from Manuscript: (Page 17) …… Supplementary file 1f showed the number and size of genomic regions of each method and the overlapping among these methods ……

b. Second, please provide a way to evaluate whether differences in enrichment between the methods are statistically meaningful.

We thank the reviewers for the advice to statistically evaluate the difference of heritability enrichment to show the advantage of SpecVar. The standard error of heritability enrichment was estimated by block jackknife, which allowed us to conduct a t-test for the significance of the heritability enrichment difference. For example, SpecVar gave LDL’s heritability enrichment in the “right lobe of liver” to be 678.91 with a standard error of 392.43 and ARE gave a heritability enrichment of 113.34 with a standard error of 30.45. We conducted Welch's t-test and revealed the p-value of their difference to be 7.0e-4. We computed all the p-values of difference between SpecVar and the other four methods and showed them in the figure of our manuscript.

Excerpt from Manuscript: (Page 5) …… First, we showed that SpecVar could achieve higher heritability enrichment in the relevant tissues than other methods (Supplementary file 1b). Taking LDL for example, SpecVar obtained a heritability enrichment of 678.91 in the “right lobe of liver”, while ARE, SAP, AAP, and SEG gave heritability enrichment of 113.34, -42.09, 50.95, 4.47, respectively. We conducted Welch’s t-test to assess the significance of the difference between SpecVar and other methods and found heritability enrichment of SpecVar was significantly higher than ARE (p=6.9×10−4), SAP (p=1.4×10−4), AAP (p=3.4×10−4), and SEG (p=2.1×10−4) (Figure 2a). For total cholesterol, SpecVar also gave significantly higher heritability enrichment in “right lobe of liver” than ARE (p=5.7×10−4), SAP (p=4.4×10−5), AAP (p=1.6×10−4), and SEG (p=7.7×10−5) (Figure 2b). For educational attainment and cognitive performance, they were relevant to brain tissues: “frontal cortex”, “cerebellum”, “caudate nucleus”, “Ammon’s horn” and “putamen”. SpecVar obtained the highest averaged heritability enrichment in brain tissues among these methods (Figure 2—figure supplement 1a, b). In the “frontal cortex”, SpecVar had significantly higher heritability enrichment than ARE (p=1.2×10−5), SAP (p=2.0×10−6), AAP (p=3.0×10−6), and SEG (p=3.0×10−6) for educational attainment (Figure 2c). And for cognitive performance in “frontal cortex”, SpecVar also had significantly higher heritability enrichment than ARE (p=9.0×10−6), SAP (p=2.0×10−6), AAP (p=1.0×10−6), and SEG (p=1.0×10−6) (Figure 2d). For brain shape, SpecVar obtained a significantly higher heritability enrichment in its relevant context “CNCC” than the other four methods (ARE p=5.9×10−4, SAP p=6.7×10−4, AAP p=7.5×10−4, SEG p=8.1×10−5, Figure 2e). For facial morphology, SpecVar also gave a much higher heritability enrichment in “CNCC” than the other four methods (ARE p=9.0×10−6, SAP p=1.0×10−6, AAP p=7.0×10−6, SEG p=1.0×10−6, Figure 2e) ……

c. Fitting a joint LDSC model with multiple annotation sets would provide a more rigorous assessment of the performance of SpecVar relative to the other approaches.

We have followed the reviewer’s suggestions and fitted a joint LDSC model with multiple annotation sets. This provided a more rigorous assessment of the performance of SpecVar relative to the other approaches. In detail, we had 77 genomic region sets for SpecVar, ARE, SAP, AAP, and SEG. We pooled them together to form a genome partition of 385 genomic region sets. Then we fitted the stratified LDSC model with this pooled genome partition and obtained heritability enrichments and R scores.

First, we compared the heritability enrichment of five methods in relevant tissues. From Figure 2-figure supplement 2, we could see that for all six phenotypes, SpecVar achieved higher heritability enrichment than other methods. We also used Welch's t-test to evaluate the significance of the heritability difference between SpecVar and other methods and found SpecVar’s heritability enrichments were significantly higher than that of other methods (Figure 2-figure supplement 2).

Second, we used the pooled genome partition to compare three specificity-based methods to show if they could identify relevant tissue for six phenotypes. As in our manuscript, we used heritability enrichment and their p-values to define the R score. We found SpecVar can still identify relevant tissues for these six phenotypes (Author response image 1). For example, SpecVar ranked the “right lobe of liver” to be the most relevant tissue for both LDL and total cholesterol. SAP identified the “Hela-S3” cell line for LDL and total cholesterol. SEG identified “HepG2” as the most relevant tissue for LDL and total cholesterol. And for educational attainment and cognitive performance, SpecVar obtained brain tissues to be top-ranked tissues, such as “the frontal cortex”, “Ammon’s horn”, and “putamen” in the top 5 tissues for educational attainment and “Ammon’s horn”, “putamen” in the top 5 tissues for cognitive performance. But for SAP, it only identified “Ammon’s horn” in the top 5 tissues for educational attainment and identified no brain tissues for cognitive performance. And for SEG, no tissues in the top 5 tissues for educational attainment and cognitive performance were from the brain. Finally, for brain shape and facial morphology, SpecVar could still rank “CNCC” to be the most relevant tissue but SAP and SEG couldn’t identify “CNCC”.

In summary, these results showed SpecVar could improve heritability enrichment and identify relevant tissues for GWAS summary statistics. We have discussed these results in our manuscript.

Excerpt from Manuscript: (Page 12) …… The improved heritability enrichment was also observed when we pooled regulatory categories of all the methods together to fit stratified LDSC and re-estimated the heritability enrichment of each context for each method (Figure 2—figure supplement 2) ……

4. Y-axes for bar plots in Figure 5 should start at 0. The floating y-axis origin is highly misleading.

We are sorry for the misleading y-axis origin, and we have changed them to be from 0 in Figure 5.

5. The approach to defining context-specific REs based on overlap (>50% of bases overlapping for cross-group comparisons, and >60% of bases overlapping for within-group comparisons) seems quite stringent and likely excludes a lot of lineage-specific REs. Calculating enrichments for "group-specific" regulatory networks, where REs are considered group-specific if they do not overlap any of the other 35 groups resulting from the hierarchical clustering (ignoring whether an RE overlaps REs from other contexts within the same group), might provide more biologically relevant region sets.

We agree that our approach to defining context-specific REs is stringent and we follow the reviewer’s suggestion to define specificity at the “group” level. In detail, we have classified the 77 regulatory networks into 36 groups in our manuscript. First, we pooled the REs in the same groups together to form 36 group-level RE sets. Then we defined a RE to be a group-specific RE if it was not overlapped with REs in other groups. This procedure would form 36 sets of group-specific REs. Finally, we fitted the stratified LDSC model with these 36 sets of group-specific genomic regions and obtained heritability enrichment, p-value, and R score for each group.

Figure 3-figure supplement 5 shows the top 5 groups of tissues ranked by their R scores. We can see that SpecVar can also reveal relevant tissues. For LDL and total cholesterol, SpecVar identified Liver to be the most relevant tissue. For educational attainment and cognitive performance, SpecVar revealed brain tissues to be the most relevant tissues. And for brain shape and facial morphology, our method found CNCC to be the most relevant tissue (Figure 3—figure supplement 5a-f).

We also found group based SpecVar gives a larger difference between relevant tissue and irrelevant tissues. We used the fold change between the first ranked relevant group/tissue and the second ranked relevant group/tissue to evaluate the difference between relevant tissue and irrelevant tissues. For LDL, group-based SpecVar gave a fold change of 9.09 and the original SpecVar gave 4.78. For total cholesterol, group-based SpecVar gave a fold change of 8.93 and the original SpecVar gave 3.30. For educational attainment, group-based SpecVar gave a fold change of 14.46 and the original SpecVar gave 1.82. For cognitive performance, group-based SpecVar gave a fold change of 3.72 and the original SpecVar gave 2.01. For facial morphology, group-based SpecVar gave a fold change of 2.60 and the original SpecVar gave 3.56. For brain shape, group-based SpecVar gave a fold change of 1.57 and the original SpecVar gave 1.04 (Figure 3—figure supplement 5g). This showed that group-based SpecVar could give more biologically relevant region sets in most situations.

In summary, we used 36 groups of 77 contexts to build group-based SpecVar and the results also showed that SpecVar can identify the accurate relevant group of tissues. Group-based SpecVar gave larger differences between relevant groups and irrelevant groups with the price of resolution. For example, group-based SpecVar could only tell us educational attainment and cognitive performance were relevant to brain tissues and couldn’t prioritize more specific relevant parts of the brain. And the original SpecVar could reveal these two phenotypes were more relevant to the “frontal cortex”. We thank the reviewer for this suggestion again and we have discussed these results in our revised manuscript.

Excerpt from Manuscript: (Page 13) …… Second, we could identify relevant organs or groups of tissues. For example, our 77 human contexts can be hieratically clustered into 36 groups. We pooled the REs in the same groups together to form 36 group-level RE sets. Then we defined a RE to be a group-specific RE if it was not overlapped with REs in other groups. This procedure would form 36 sets of group-specific REs. Finally, we fitted the stratified LDSC model with these 36 sets of group-specific regulatory categories and obtained heritability enrichment, P-value, and R score for each group by SpecVar. We tested the group-based SpecVar on our six phenotypes. For LDL and total cholesterol, SpecVar identified “Liver” tissues to be the most relevant tissue (Figure 3—figure supplement 5a, b). For educational attainment and cognitive performance, SpecVar revealed “brain” tissues to be the most relevant tissues (Figure 3—figure supplement 5c, d). And for brain shape and facial morphology, SpecVar found “CNCC” to be the most relevant tissue (Figure 3—figure supplement 5e, f) ……

Reviewer #2 (Recommendations for the authors):Specific comments– The authors only show the performance of the SpecVar method on 6 traits, however, it is not clear whether these are representative of all of the traits in UKBiobank. For traits with fewer SNP associations and lower sample numbers it appears that LDSC-SAP produces much high trait relevance scores, however, the authors use different tissues in each method so it may not be a fair comparison.

We are sorry that we did not clarify the reason for choosing these 6 traits clear. Generally, it is a difficult task to select representative phenotypes with confirmed knowledge of relevant tissues as gold-standard for validation. We use the following three conditions to select gold standard traits:

1. These traits should be representative and from different aspects. Here LDL and total cholesterol are lipid molecular phenotypes. Educational attainment and cognitive performance are human intelligential traits by intelligence test. Brain shape and facial morphology are human craniofacial traits from pictures of human faces and MRI of human brains.

2. These traits should be well studied to know the relevant tissues. We have prior knowledge about the relevant tissue of these six phenotypes to some extent. Lipid phenotypes are associated with the liver for its key role in lipid metabolism (Nguyen et al., 2008); human intelligential phenotypes are associated with brain tissues (Goriounova and Mansvelder, 2019); facial morphology and brain shape had shared heritability in cranial neural crest cells (CNCC) (Naqvi et al., 2021).

3. These traits should have GWAS with enough power, which means they should have enough sample size and significant SNP associations. The sample size and number of significant SNPs are large for these six phenotypes’ GWAS.

We are also sorry we did not make the tissues used for each phenotype clear. For every phenotype, SpecVar computes its relevance scores, p-values, and q-values to all 77 contexts. And we use relevance scores and q-values to select relevant tissues. In Figure 3 of our manuscript, we only show the top 5 tissues among 77 tissues ranked by relevance scores of each method. The full list of 77 contexts with their relevance score were provided in the Supplementary file 1c.

– The authors use the Pearson correlation coefficient to evaluate the performance of the SpecVar method, however, they should also consider other nonlinear metrics.

We thank the reviewer’s suggestion, and we used mutual information (MI) as an additional nonlinear metric to evaluate the accuracy of estimating phenotypic correlation. As shown in Figure 5-figure supplement 1 and 2, SpecVar achieved larger MI than SAP and SEG for all phenotype pairs and highly correlated pairs in the studies of facial morphology and UKBB. We have added this new metric to our revised manuscript.

Excerpt from Manuscript: (Page 10) …… We also used the mean square error (Figure 5—figure supplement 1a, b) and mutual information (Figure 5—figure supplement 1c, d) as metrics to evaluate the performance (Methods) and SpecVar was the best among these three methods …… SpecVar’s outperformance of estimating phenotype correlation was reproduced by using mean square error (Figure 5—figure supplement 2a, b) and mutual information (Figure 5—figure supplement 2c, d) as metrics ……

– The authors should consider a comparison with other regulatory network-based heritability methods such as CoCoNet, which is based on co-regulated genes. They should also compare to other non-network-based methods.

We thank the reviewer’s suggestion to make additional comparisons with the network-based method and non-network-based method. For the network-based method, we chose CoCoNet, which was based on gene co-expression networks. And for the non-network-based method, we chose RolyPoly, which was based on regression on gene expression profiles.

First, we applied CoCoNet to our six phenotypes with 38 built-in GTEx co-expression networks. Figure 3-figure supplement 1 shows the top 5 tissues ranked by loglikelihood of CoCoNet. We could see CoCoNet identified “Breast” as the most relevant tissue for LDL and “Brain_other” as the most relevant tissue for total cholesterol. “Breast” was the most relevant tissue for educational attainment and “Stomach” was the most relevant tissue for cognitive performance. Since there was no CNCC sample in GTEx, CoCoNet revealed “Prostate” as the most relevant tissue for brain shape and facial morphology. These results seemed less reasonable than SpecVar because CoCoNet didn’t identify liver tissue for LDL and total cholesterol and brain tissues for educational attainment and cognitive performance.

Then we fitted the RolyPoly model with gene expression profiles of our 77 human contexts and applied it to GWAS used in our paper. Since we didn’t have β, standard error in the summary statistics of brain shape, we only applied RolyPoly to LDL, total cholesterol, educational attainment, cognitive performance, and facial morphology. Figure 3-figure supplement 2 shows the top 5 tissues ranked by -log(P-value) of RolyPoly. RolyPoly prioritized the “HepG2” cell line as the most relevant tissue for LDL and total cholesterol. HepG2 cells are nontumorigenic cells with high proliferation rates and epithelial-like morphology that perform many differentiated hepatic functions. For educational attainment, RolyPoly didn’t identify the 5 brain tissues as the top-ranked tissue and only included “fetal spinal cord” in the top 5 relevant tissues. For cognitive performance, there were no brain associated tissues in the top 5 tissues. And for facial morphology, RolyPoly also failed to identify “CNCC” as relevant tissue.

This comparison showed the strength of integrating both gene expression and chromatin accessibility into the regulatory network, which was the main contribution of SpecVar. We thank the reviewer for the suggestion of additional comparisons again and we have added this comparison to our revised manuscript.

– The highlighted example SNP-associated network for the FOXC2 variants is interesting, however, the authors should demonstrate whether there are chromatin interactions (HiC or HiChIP) in brain tissues linking these variants in ATAC-seq peaks to the FOXC2 promoter. It would also be helpful to highlight another example using distinct UKBB phenotype and tissue datasets.

We thank the reviewer’s interest in our example of FOXC2 to show that SNPs are located in the RE to regulate gene expression. We agree that we should check if there are chromatin interactions in brain tissues linking these variants in ATAC-seq peaks to the FOXC2 promoter as additional evidence of this SNP-associated regulation. To do this, we downloaded the HiChIP loops of brain tissue from the recently published HiChIP database (HiChIPdb) (Zeng et al., 2022). We searched the loops of FOXC2 and found there were 16 HiChIP loops linking REs and promoter of FOXC2. And we found this RE was linked by one of these loops to the FOXC2 promoter (Figure 4), which showed that the SNP-associated regulation of FOXC2 was supported by independent HiChIP data.

It is a good idea to show another example using distinct UKBB phenotypes and tissues. Here we used SH2B1 as an example (Figure 6c). In our manuscript, “body mass index” and “leg fat-free mass (right)” were revealed to be correlated by SpecVar’s relevance correlation. SpecVar further revealed that these two phenotypes were correlated because they were both relevant to the “frontal cortex”. SH2B1 was a target gene in the shared SNP associated regulatory network in the “frontal cortex” of “body mass index” and “leg fat-free mass (right)”. SH2B1 enhances leptin signaling and leptin’s anti-obesity action, which is associated with the regulation of energy balance, body weight, and glucose metabolism (Rui, 2014). We found one common significant SNP located in the specific REs of the “frontal cortex” at the 90k bp downstream of SH2B1. Even though this RE was near the promoter of NFATC2IP, our regulatory network revealed that it regulated the expression of SH2B1, which indicated the common SNP might regulate the expression of SH2B1 to influence phenotypes of “body mass index” and “leg fat-free mass (right)”. We also checked the HiChIP loops of brain tissues in HiChIPdb and found this SNP-associated regulation was supported by a HiChIP loop in brain tissues.

We thank the reviewer’s advice again to add more evidence to our SNP-associated regulation and another example of shared SNP-associated regulation for two UKBB phenotypes. We have added the additional evidence and examples to our revised manuscript.

Excerpt from Manuscript: (Page 9) …… We checked the loops in the database of HiChIP (Zeng et al., 2022) and found this SNP-associated regulation of FOXC2 was supported by a HiChIP loop in brain tissues to link this SNP locus and FOXC2 promoter ……

(Page 11) …… For example, in the brain, SH2B1 enhances leptin signaling and leptin’s anti-obesity action, which is associated with the regulation of energy balance, body weight, and glucose metabolism (Rui, 2014). We found one common significant SNP of these two phenotypes was located in the specific REs of the “frontal cortex” at the 90k downstream of SH2B1. Even though this RE was near the promoter of NFATC2IP, SpecVar revealed that it regulated the expression of SH2B1, which was supported by a HiChIP loop in brain tissues to associate the SNP-associated RE and promoter of SH2B1 (Figure 6c). These results indicated that one shared SNP was located in the “frontal cortex” specific RE and might regulate the expression of SH2B1 to influence two phenotypes: “body mass index” and “leg fat-free mass (right)” ……

– It will be more interesting to adapt this to paired multimodal single-cell data if possible or expand on this in the Discussion.

We thank the reviewer’s suggestion, and we agree that adapting our idea to single cell data will give us a higher resolution of relevant contexts at the cell type level, which is promising in the future. We have discussed this idea in our revision.

Excerpt from Manuscript: (Page 13) …… Third, we can identify relevant contexts at the cell type level since paired multimodal single-cell data, such as single cell RNA-seq and single cell ATAC-seq data, are increasing in recent years (Han et al., 2020). The paired multi-omics data, which means single cell data profiled from the same context, enable us to identify cell types and infer regulatory networks for these cell types (Duren et al., 2018; Zeng et al., 2019). It will be promising to construct an atlas of context-specific regulatory networks at cell type level and build SpecVar model based on these cell-type-specific regulatory networks. This extension of SpecVar to single cell level holds the promise to identify more detailed relevant cell types for given phenotypes ……

– Lastly, LDSC-based methods may not perform well on admixed populations, thus some discussion on how this could be adapted using covariate-adjusted approaches (e.g. cov-LDSC) would be helpful.

We agree that it is important to consider the admixed population and our model should be better if we consider the mixed population. We discussed this potential combination of our idea and cov-LDSC in the “Discussion” section of our manuscript.

Excerpt from Manuscript: (Page 14) …… On the other hand, SpecVar is based on stratified LDSC, which may not perform well on admixed populations. Thus, considering the effect of mixed ancestries will help SpecVar handle more circumstances. For example, Luo. et al. has developed cov-LDSC to adjust the LDSC model to be suitable for admixed populations (Luo et al., 2021). Building SpecVar model based on cov-LDSC will be promising to perform well on GWAS with mixed populations ……

Reviewer #3 (Recommendations for the authors):The study would benefit from investigating the following questions:Which part of the new annotation drives the extreme heritability enrichment from SpecVar, compared to other LDSC-based methods?

We agree that SpecVar gives much higher heritability enrichment than the other methods in the relevant tissues. The heritability enrichment of LDL in the “right lobe of liver” was 678.91. This was an extremely high heritability enrichment because the mean and median of heritability enrichment for six phenotypes in 77 tissues were only 15.12 and 6.55, respectively.

First, we followed the reviewer’s suggestion and provided a standard error of each heritability enrichment in our revision (Figure 2). For example, the standard error of LDL’s heritability enrichment in “right lobe of liver” was 392.43, which was larger than the standard error of ARE (30.45), SAP (58.56), AAP (9.96), and SEG (0.83). For the other five traits, SpecVar also gave higher standard error than the other methods, which might be caused by the smaller proportion of the genome covered by SpecVar than the other methods. We could use the standard error to conduct a t-test to assess the heritability enrichment difference between SpecVar and other methods. We found SpecVar had significantly higher heritability enrichment than other methods. Taking LDL in the “right lobe of liver” as an example, SpecVar gave significantly higher heritability than ARE (P=6.9e-4), SAP (P=1.4e-4), AAP (3.4e-4), and SEG (2.1e-4). We also provided the p-values of the heritability enrichment comparison of our manuscript (Figure 2).

Second, this is true that the main difference between SpecVar and other methods is the annotated genomic regions. We totally agree that it is important to understand the difference between the SpecVar annotated SNPs and those from other methods and to understand where the extra heritability enrichment comes from. For LDL in the “right lobe of liver”, if we only used annotation of gene expression (SEG), the heritability enrichment was 4.47, and if we only used annotation of chromatin accessibility (AAP), the heritability enrichment was 50.95. We first built new annotation by integrating gene expression and chromatin accessibility into the regulatory network (ARE), which gave a heritability of 113.34. So, the SpecVar-specific annotation gave a 25-fold improvement in heritability enrichment. And SpecVar further considered the specificity of ARE to narrow down the annotated genomic regions, which gave another 6-fold improvement of heritability on ARE. From this example, we can hypothesize that the regulatory network (SpecVar-specific annotation) may play the main role in improving the heritability enrichment, and specificity (intersection narrowed down) plays a secondary role. To validate this hypothesis, we conducted the ablation analysis. First, we listed main components of SpecVar (Figure 2—figure supplement 1C) to illustrate the relationship and differences of the five methods: SEG was the combination of gene expression and specificity; AAP was only from chromatin accessibility; SAP was the combination of chromatin accessibility and specificity; ARE integrated gene expression and chromatin accessibility; SpecVar considered gene expression, chromatin accessibility, and specificity. Then we compared SpecVar to ARE to show the effect of specificity, compared SpecVar to SAP to show the effect of gene expression annotations, and compared SpecVar to SEG to show the effect of chromatin accessibility (Figure 2—figure supplement 1D). We conducted these comparisons on six phenotypes and computed fold change of heritability enrichment to measure the effect. We found that the annotation from chromatin accessibility gave the largest effect to improve heritability enrichment (Figure 2—figure supplement 1E). This meant that the regulatory networks of SpecVar, which integrated chromatin accessibility with gene expression data, played a more important role in heritability enrichment. And specificity could also improve heritability enrichment.

We thank the reviewer’s suggestion to help us analyze the contribution of each part of our method. We have added this analysis to our manuscript.

Excerpt from Manuscript: (Page 5) …… First, we showed that SpecVar could achieve higher heritability enrichment in the relevant tissues than other methods (Supplementary file 1b). Taking LDL for example, SpecVar obtained a heritability enrichment of 678.91 in the “right lobe of liver”, while ARE, SAP, AAP, and SEG gave heritability enrichment of 113.34, -42.09, 50.95, 4.47, respectively. We conducted Welch’s t-test to assess the significance of the difference between SpecVar and other methods and found heritability enrichment of SpecVar was significantly higher than ARE (p=6.9×10−4), SAP (p=1.4×10−4), AAP (p=3.4×10−4), and SEG (p=2.1×10−4) (Figure 2a). For total cholesterol, SpecVar also gave significantly higher heritability enrichment in “right lobe of liver” than ARE (p=5.7×10−4), SAP (p=4.4×10−5), AAP (p=1.6×10−4), and SEG (p=7.7×10−5) (Figure 2b). For educational attainment and cognitive performance, they were relevant to brain tissues: “frontal cortex”, “cerebellum”, “caudate nucleus”, “Ammon’s horn” and “putamen”. SpecVar obtained the highest averaged heritability enrichment in brain tissues among these methods (Figure 2—figure supplement 1a, b). In the “frontal cortex”, SpecVar had significantly higher heritability enrichment than ARE (p=1.2×10−5), SAP (p=2.0×10−6), AAP (p=3.0×10−6), and SEG (p=3.0×10−6) for educational attainment (Figure 2c). And for cognitive performance in “frontal cortex”, SpecVar also had significantly higher heritability enrichment than ARE (p=9.0×10−6), SAP (p=2.0×10−6), AAP (p=1.0×10−6), and SEG (p=1.0×10−6) (Figure 2d). For brain shape, SpecVar obtained a significantly higher heritability enrichment in its relevant context “CNCC” than the other four methods (ARE p=5.9×10−4, SAP p=6.7×10−4, AAP p=7.5×10−4, SEG p=8.1×10−5, Figure 2e). For facial morphology, SpecVar also gave a much higher heritability enrichment in “CNCC” than the other four methods (ARE p=9.0×10−6, SAP p=1.0×10−6, AAP p=7.0×10−6, SEG p=1.0×10−6, Figure 2e) ……

(Page 6) …… To explore the heritability enrichment improvement of SpecVar, we conducted ablation analysis to study the contribution of two important parts of SpecVar: (i) regulatory network by integrating gene expression and chromatin accessibility data, and (ii) specificity by comparing with other contexts. Figure 2—figure supplement 1c shows the relationship and difference of the five methods: SEG is the combination of gene expression and specificity; AAP is only from chromatin accessibility; SAP is the combination of chromatin accessibility and specificity; ARE integrates gene expression and chromatin accessibility; and SpecVar considers integration of gene expression and chromatin accessibility, and specificity. To show the effect of the regulatory network in heritability enrichment, we compared SpecVar with SAP and showed the effect of gene expression data. We compared SpecVar with SEG and showed the effect of chromatin accessibility data. We compared SpecVar with ARE and showed the contribution of specificity (Figure 2—figure supplement 1d). To quantify the effect of each component, we computed the fold change of different methods’ heritability enrichments. We found chromatin accessibility, which was part of the regulatory network, showed the highest effect in improving heritability enrichments for all six phenotypes. This is consistent with the fact that most genetic variants are located in the non-coding regulatory regions (Claussnitzer et al., 2015; Kumar et al., 2012; Smemo et al., 2014) and accessibility gives the direct functional evidence for genetic variants. The specificity in SpecVar also contributed at least 4-fold improvement for heritability enrichment (Figure 2—figure supplement 1e) ……

How to use the R score for a formal hypothesis test, rather than subjectively picking a few top-ranked tissues for the trait? How to evaluate the false positive rate for the test?

We thank the reviewer to raise the question about the formal hypothesis test and false positive rate of R scores. Here we used the block jackknife stratagem to estimate standard errors of R scores and used these standard errors to calculate z scores, p-values, and false discovery rates. The block jackknife method was also used in the stratified LDSC paper (Finucane et al., 2015).

Given a GWAS summary statistics, we had the following steps to conduct block jackknife to estimate the standard error of its R score to a context:

1) For this context, we divided the context-specific RE into 100 folds.

2) One sub-sample was generated by removing one of the 100 folds and we generated 100 sub-samples of the context-specific REs.

3) The 100 sub-samples of specific REs would form a new genome partition for fitting stratified LDSC.

4) For each sub-sample, we obtained heritability enrichment, p-value, and R score by SpecVar.

5) With the 100 background R scores of the 100 sub-samples, we could estimate the standard error (SD) of R scores for this context.

5) We computed the z score of R score: Z=R/SD∼N(0,1), and estimated the p-value and FDR q-value.

The R scores and their FDR q-values could be used to select relevant tissues. We used R score ≥100 and FDR ≤0.01 as the threshold to pick up relevant tissues for a phenotype. In our paper, the relevant tissues for six phenotypes were listed within Table 1.

How to explain the inconsistent findings with previous studies, e.g., the association of the liver to LDL?

We agree that there is some difference between our results and the LDSC-SEG paper. In the LDSC-SEG paper, there were three liver tissues that were from the GTEx dataset. And in their Figure 2, the -log(P-value) of the three liver tissues’ relevance to LDL was from 3.3-4.4. In our implementation, the -log(P-value) of the “right lobe of liver” to LDL was 4.1. So, there was minor difference between our implementation and the LDSC-SEG paper.

The minor difference between our results and the LDSC-SEG paper may be caused by the expression profiles of different tissues. The expression data of the LDSC-SEG paper was mainly from GTEx and our paper included expression data of more tissues from ENCODE, such as HepG2, adrenal gland, thoracic aorta, etc. Specifically expressed genes were different when we used different tissues for comparison. For example, for the liver tissue, 30% of the specific genes in our paper were different from the specific genes in the LDSC-SEG paper. This will make different heritability enrichment and p-value of liver tissue. And the extra tissues included in our paper also made it possible to select other reasonable samples to be relevant to LDL, such as “HepG2”, which are nontumorigenic cells with high proliferation rates and epithelial-like morphology that perform many differentiated hepatic functions. This made the “right lobe of liver” fail to be ranked to be top 5. Noting that there were expression data of 27 samples in our 77 contexts which were also from GTEx, we could fit LDSC-SEG with these 27 samples. And we found if we only used these 27 GTEx samples, the “right lobe of liver” came to be most relevant tissue (P=5.1e-6, Q=1.2e-4).

Are the GWAS signal in context-specific RE colocalised with context-specific eQTL signals?

We first thank the reviewer’s precise summary of our effort to highlight an example where SpecVar facilitates the interpretation of GWAS signals near FOXC2. We totally agree that more work can be done to consolidate the SNP-associated regulation of FOXC2.

eQTL is a good resource to validate the SNP association regulation. We collected eQTL of all brain tissues from GTEx and obtain 1,431,702 SNP-gene pairs. For FOXC2, we found two eQTLs (chr16:86511797-FOXC2 and chr16:86512080-FOXC2) and they were not in the locus we identified. This inconsistency might be caused by the context since our SNP-associated regulation was identified in CNCC and there are no eQTL data for CNCC. Then we tried another way to consolidate the SNP-associated regulation of FOXC2 by checking if there was a chromatin contact between the SNP locus and FOXC2 promoter. To do this, we downloaded the HiChIP loops of brain tissue from the recently published HiChIP database (HiChIPdb) (Zeng et al., 2022). We searched the loops of FOXC2 and found there were 16 HiChIP loops linking REs and promoter of FOXC2. And we found this RE was linked by one of these loops to the FOXC2 promoter (Figure 4b), which showed that the SNP-associated regulation of FOXC2 was supported by independent HiChIP data. On the other hand, the top GWAS signal in this locus was indeed on the left of the CNCC-specific RE, which meant that they were not active in CNCC. As we know, brain shape is a human complex trait, and it will be relevant to more than one tissues. Here the SNPs of brain shape were only studied in CNCC, and they might also be active in other tissues or at other developmental stages. Including more contexts and cell types may be promising to understand all the genetic variants of this complex trait.

We can used eQTL data to validate SNP-associated regulations of educational attainment, cognitive performance, LDL, and total cholesterol because there are eQTLs of brain and liver in GTEx. For educational attainment, SpecVar revealed that 7,611 SNP-TG pairs in its SNP associated regulatory network of “frontal cortex” and we found 3,693 SNPs (40.1%) of these SNP-TG pairs were also SNPs of eQTL in brain tissues. Among the 2,862 SNPs, 788 SNPs (10.4%) had the same TGs with the eGenes in the eQTL database (hypergeometry test p-value=7.0e-121). For cognitive performance, there were 7,494 SNP-TG pairs in its SNP associated regulatory network of “frontal cortex”. And 3,569 of them (47.6%) were also SNPs of eQTL in brain tissues. SpecVar further revealed that 988 SNPs (13.2%) had the same TGs with the eGenes in the eQTL database (p-value=3.2e-224). For LDL, there were 556 SNP-TGs pairs in its SNP associated regulatory network of “right lobe of liver”. We collected 323,428 eQTL of liver tissue from GTEx and found 45 of the SNP-TG pairs (8.1%) were also SNPs of eQTL. Two of the SNPs had the same TGs with the eGenes in the eQTL database (p-value=0.05). For total cholesterol, there were 461 SNP-TGs pairs in its SNP associated regulatory network of “right lobe of liver”. We found 16 of the SNP-TG pairs (3.5%) were also SNPs of eQTL. There were two of the SNPs that had the same TGs with the eGenes in the eQTL database (p-value=0.03).

We thank the reviewer again for the suggestion to add more evidence to support SpecVar-revealed SNP-associated regulations. We have added the HiChIP loop of FOXC2 and eQTL analysis to our revised manuscript.

Excerpt from Manuscript: (Page 9) …… We checked the loops in the database of HiChIP (Zeng et al., 2022) and found this SNP-associated regulation of FOXC2 was supported by a HiChIP loop in brain tissues to link this SNP locus and FOXC2 promoter……

(Page 9) …… The SNP-associated regulatory can facilitate the fine mapping of GWAS signals. Since there are eQTL data of the liver and brain tissues in the GTEx database, we collected the significant SNP-gene pairs of liver and brain tissues from GTEx to validate the identified SNP-associated regulations. For educational attainment, SpecVar revealed 7,611 SNP-TG pairs in its SNP associated regulatory network of “frontal cortex” and we found 3,693 SNPs (40.1%) of these SNP-TG pairs were also SNPs of eQTL in brain tissues. Among the 2,862 SNPs, 788 SNPs (10.4%) had the same TGs with the eGenes in the eQTL database (hypergeometry test, p=7.0×10−121). For cognitive performance, there were 7,494 SNP-TG pairs in its SNP associated regulatory network of “frontal cortex”. And 3,569 of them (47.6%) were also SNPs of eQTL in brain tissues. SpecVar further revealed that 988 SNPs (13.2%) had the same TGs with the eGenes in the eQTL database (p=3.2×10−224). Then we collected 323,428 eQTL of liver tissues from GTEx. For LDL, there were 556 SNP-TGs pairs in its SNP associated regulatory network of “right lobe of liver” and we found 45 of the SNP-TG pairs (8.1%) were also SNPs of eQTL. Two of the SNPs had the same TGs with the eGenes in the eQTL database (p=5.0×10−2). For total cholesterol, there were 461 SNP-TGs pairs in its SNP associated regulatory network of “right lobe of liver”. We found 16 of the SNP-TG pairs (3.5%) were also SNPs of eQTL. There were two of the SNPs that had the same TGs with the eGenes in the eQTL database (p=3.0×10−2) ……

How to link relevance score correlation across tissues to shared heritability between traits?

We thank the reviewer for the summary about SpecVar’s relevance correlation and we agree that it is interesting to study the observed phenotypic correlation by common genetic factors acting in the shared tissues/cell types/pathways/regulatory networks between traits. SpecVar could utilize the correlation of relevance scores to 77 contexts to estimate relevance correlation. Furthermore, SpecVar revealed two phenotypes were correlated because they were relevant to the same contexts. And in the common relevant context, SpecVar could identify shared SNP-associated regulatory network underlying their relevance correlation.

In the application to the UKBB dataset, SpecVar gave us the relevance correlation, the common relevant tissues, and the shared SNP-associated regulatory network among UKBB phenotypes (Supplementary file 1e). For example, SpecVar found “body mass index” and “leg fat-free mass (right)” were correlated with a relevance correlation of 0.602. SpecVar further revealed that these two phenotypes were correlated because they were both relevant to the “frontal cortex” (Figure 6a). Body mass index has been reported to be related to frontal cortex development (Laurent et al., 2020) and relevant to the reduced and thin frontal cortex (Islam et al., 2018; Shaw et al., 2018). Obesity and fat accumulation are also revealed to be associated with the frontal cortex (Gluck et al., 2017; Kakoschke et al., 2019). SpecVar further extracted these two phenotypes’ SNP-associated regulatory networks in the “frontal cortex” and found their SNP-associated networks were significantly overlapped. The significant overlap was observed at SNP, RE, TG, and TF levels: p=8.2×10−63 for SNPs, p=1.4×10−47 for REs, p=6.0×10−25 for TGs, and p=8.2×10−25 for TFs (Figure 6b). The shared regulatory network was involved with body weight and obesity. For example, in the brain, *SH2B1* enhances leptin signaling and leptin’s anti-obesity action, which is associated with the regulation of energy balance, body weight, and glucose metabolism (Rui, 2014). We next studied the detailed SNP-associated regulation of SH2B1 (Figure 6c). We found that one common significant SNP was located in the specific REs of the “frontal cortex” at the 90k bp downstream of SH2B1. Even though this RE was near the promoter of NFATC2IP, our regulatory network revealed that it regulated the expression of SH2B1, which indicated the common SNP may regulate the expression of SH2B1 to influence phenotypes of “body mass index” and “leg fat-free mass (right)”. We also checked the loops of brain tissues in HiChIPdb and found this SNP-associated regulation was also supported by a HiChIP loop in brain tissues.

We have added the advantage of SpecVar to reveal the common genetic factors acting in the common relevant tissues and shared regulatory networks underlying the correlation between phenotypes to our revision.

Reference

Feng, Z. Y., Duren, Z. N., Xiong, Z. Y., Wang, S. J., Liu, F., Wong, W. H., and Wang, Y. (2021). hReg-CNCC reconstructs a regulatory network in human cranial neural crest cells and annotates variants in a developmental context. *Communications Biology*, *4*(1). https://doi.org/ARTN 442

10.1038/s42003-021-01970-0

Finucane, H. K., Bulik-Sullivan, B., Gusev, A., Trynka, G., Reshef, Y., Loh, P. R., Anttila, V., Xu, H., Zang, C. Z., Farh, K., Ripke, S., Day, F. R., Purcell, S., Stahl, E., Lindstrom, S., Perry, J. R. B., Okada, Y., Raychaudhuri, S., Daly, M. J.,... Consortium, R. (2015). Partitioning heritability by functional annotation using genome-wide association summary statistics. *Nature Genetics*, *47*(11), 1228-+. https://doi.org/10.1038/ng.3404

Gluck, M. E., Viswanath, P., and Stinson, E. J. (2017). Obesity, Appetite, and the Prefrontal Cortex. *Curr Obes Rep*, *6*(4), 380-388. https://doi.org/10.1007/s13679-017-0289-0

Goriounova, N. A., and Mansvelder, H. D. (2019). Genes, Cells and Brain Areas of Intelligence. *Front Hum Neurosci*, *13*, 44. https://doi.org/10.3389/fnhum.2019.00044

Islam, A. H., Metcalfe, A. W. S., MacIntosh, B. J., Korczak, D. J., and Goldstein, B. I. (2018). Greater body mass index is associated with reduced frontal cortical volumes among adolescents with bipolar disorder. *Journal of Psychiatry and Neuroscience*, *43*(2), 120-130. https://doi.org/10.1503/jpn.170041

Kakoschke, N., Lorenzetti, V., Caeyenberghs, K., and Verdejo-Garcia, A. (2019). Impulsivity and body fat accumulation are linked to cortical and subcortical brain volumes among adolescents and adults. *Scientific Reports*, *9*. https://doi.org/ARTN 2580

10.1038/s41598-019-38846-7

Laurent, J. S., Watts, R., Adise, S., Allgaier, N., Chaarani, B., Garavan, H., Potter, A., and Mackey, S. (2020). Associations Among Body Mass Index, Cortical Thickness, and Executive Function in Children. *JAMA Pediatrics*, *174*(2), 170-177. https://doi.org/10.1001/jamapediatrics.2019.4708

Li, L., Wang, Y., Torkelson, J. L., Shankar, G., Pattison, J. M., Zhen, H. H., Fang, F., Duren, Z., Xin, J., Gaddam, S., Melo, S. P., Piekos, S. N., Li, J., Liaw, E. J., Chen, L., Li, R., Wernig, M., Wong, W. H., Chang, H. Y., and Oro, A. E. (2019). TFAP2C- and p63-Dependent Networks Sequentially Rearrange Chromatin Landscapes to Drive Human Epidermal Lineage Commitment. *Cell Stem Cell*, *24*(2), 271-284 e278. https://doi.org/10.1016/j.stem.2018.12.012

Ma, S., Chen, X., Zhu, X., Tsao, P. S., and Wong, W. H. (2022). Leveraging cell-type-specific regulatory networks to interpret genetic variants in abdominal aortic aneurysm. *Proc Natl Acad Sci U S A*, *119*(1). https://doi.org/10.1073/pnas.2115601119

Naqvi, S., Sleyp, Y., Hoskens, H., Indencleef, K., Spence, J. P., Bruffaerts, R., Radwan, A., Eller, R. J., Richmond, S., Shriver, M. D., Shaffer, J. R., Weinberg, S. M., Walsh, S., Thompson, J., Pritchard, J. K., Sunaert, S., Peeters, H., Wysocka, J., and Claes, P. (2021). Shared heritability of human face and brain shape. *Nature Genetics*. https://doi.org/10.1038/s41588-021-00827-w

Nguyen, P., Leray, V., Diez, M., Serisier, S., Le Bloc'h, J., Siliart, B., and Dumon, H. (2008). Liver lipid metabolism. *J Anim Physiol Anim Nutr (Berl)*, *92*(3), 272-283. https://doi.org/10.1111/j.1439-0396.2007.00752.x

Rui, L. (2014). SH2B1 regulation of energy balance, body weight, and glucose metabolism. *World J Diabetes*, *5*(4), 511-526. https://doi.org/10.4239/wjd.v5.i4.511

Shaw, M. E., Sachdev, P. S., Abhayaratna, W., Anstey, K. J., and Cherbuin, N. (2018). Body mass index is associated with cortical thinning with different patterns in mid- and late Life. *International Journal of Obesity*, *42*(3), 455-461. https://doi.org/10.1038/ijo.2017.254

Westra, H. J., and Franke, L. (2014). From genome to function by studying eQTLs. *Biochim Biophys Acta*, *1842*(10), 1896-1902. https://doi.org/10.1016/j.bbadis.2014.04.024

Xin, J., Hao, J., Chen, L., Zhang, T., Li, L., Chen, L., Zhao, W., Lu, X., Shi, P., and Wang, Y. (2020). ZokorDB: tissue specific regulatory network annotation for non-coding elements of plateau zokor. *Quantitative Biology*, *8*(1), 43-50. https://doi.org/10.1007/s40484-020-0195-4

Zeng, W., Liu, Q., Yin, Q., Jiang, R., and Wong, W. H. (2022). HiChIPdb: a comprehensive database of HiChIP regulatory interactions. *Nucleic Acids Res*. https://doi.org/10.1093/nar/gkac859

Zhu, X., Duren, Z., and Wong, W. H. (2021). Modeling regulatory network topology improves genome-wide analyses of complex human traits. *Nature Communications*, *12*(1), 2851. https://doi.org/10.1038/s41467-021-22588-0